# Carbenoid-involved reactions integrated with scaffold-based screening generates a Nav1.7 inhibitor
Jirong Shu[1,3], Yuwei Wang[2,3], Weijie Guo[2], Tao Liu[2], Song Cai [2,4], Taoda Shi [1,4] ✉ & Wenhao Hu [1,4]

The discovery of selective Nav1.7 inhibitors is a promising approach for developing anti-nociceptive drugs. In this study, we present a novel oxindole-based readily accessible library (OREAL), which is characterized by readily accessibility, unique chemical space, ideal drug-like properties, and structural diversity. We used a scaffold-based approach to screen the OREAL and discovered compound **C4** as a potent Nav1.7 inhibitor. The bioactivity characterization of **C4** reveals that it is a selective Nav1.7 inhibitor and effectively reverses Paclitaxel-induced neuropathic pain (PINP) in rodent models. Preliminary toxicology study shows **C4** is negative to hERG. The consistent results of molecular docking and molecular simulations further support the reasonability of the in-silico screening and show the insight of the binding mode of **C4**. Our discovery of **C4** paves the way for pushing the Nav1.7-based anti-nociceptive drugs forward to the clinic.

Voltage gated sodium channels (VGSC) are transmembrane ion channels that participate in many important physiological processes. Nine subtypes (Nav1.1 to Nav1.9) of VGSC are known in mammals, with different expression patterns in muscles, central nervous system (CNS), peripheral nervous system (PNS), and neuroendocrine cells[1,2]. Nav1.7 is mainly expressed in nociceptors and mediates human pain signal transduction and amplification[3]. In addition, many genetic studies have shown that Nav1.7 and its encoding SCN9A gene plays an important role in pain sensation[4,5]. The Nav1.7 is associated with multiple neuropathic pain disorder[6,7], and searching for its inhibitors is a pivotal approach in treating the Nav1.7-associated diseases[8].

At present, inhibitors of Nav1.7 can be divided into three categories: conventional inhibitors, sulfonamides, and acyl sulfonamides[8]. The conventional inhibitor contains various skeletal structures, including classical VGSC inhibitors lidocaine, Mexiletine, and Carbamasepine[9,10]. These classic inhibitors often have broad-spectrum VGSC channel inhibitory activity. In addition to classic inhibitors, several companies have also developed new conventional inhibitors in recent years, including AZD3161[11], Funapide[12], Vixotrigine[10], etc. these drugs showed better Nav1.7 channel selectivity. Representatives of inhibitors entered clinical trials include PF-05089771[13], GNE-616[14]. These compounds have high subtype selectivity for Nav1.7. Compared with sulfonamide derivatives, acyl sulfonamide derivatives have higher plasma concentration, which encourage researcher to apply acyl sulfonamide to replace sulfonamide[15]. PF-05241328[16], GX-585[15], and GNE-131[17] are inhibitors developed using this functional group. But few of these

molecules advanced in the clinical trial due to limited efficacy. Therefore, the discovery of novel Nav1.7 inhibitors are necessary to promote the clinic applications.

With the fact that novel molecular entities are needed to render molecular diversity for Nav1.7-based drug discovery, it is vital to come up with new screening methods. Thus, we herein reported the discovery of a new type of Nav1.7 inhibitor by using a newly developed carbenoid-involved reactions (CIRs)-based library (Fig. 1).

Drug discovery requires the interrogation of atomic interactions between ligands and pathologically relevant targets[18–20]. However, traditional libraries for high throughput screening (HTS) and virtual ligand screening (VLS) lack efficiency due to their limited accessible chemical space (<10 million compounds)[21–23]. Recently, more than 100 million readily accessible (REAL) ultra-large libraries have been developed, affording high-quality compounds for lead-to-candidate optimization[24–27]. These make-on-demand libraries have been developed by many companies or research groups, including Enamine[20] and WuXi AppTec[28], as well as Professor Ouyang's group[29]. However, ultra-large libraries docking requires extraordinarily high cost on computing resources, which is out of the capability of lots of drug discovery groups. The bottleneck has been removed by virtual synthon hierarchical enumeration screening (V-SYNTHES) strategy which uses synthon-based ligand screening to avoid costly direct screening on the fully enumerated libraries (9). The V-SYNTHES screening allows for rapid investigation of gigascale chemical space with acceptable cost. Therefore, we here borrow

[1]School of Pharmaceutical Sciences, Sun Yat-sen University, Guangzhou 510006, China. [2]Shenzhen University Health Science Center, Shenzhen 518060, China. [3]These authors contributed equally: Jirong Shu, Yuwei Wang. [4]These authors jointly supervised this work: Song Cai, Taoda Shi, Wenhao Hu ✉e-mail: shitd@mail.sysu.edu.cn

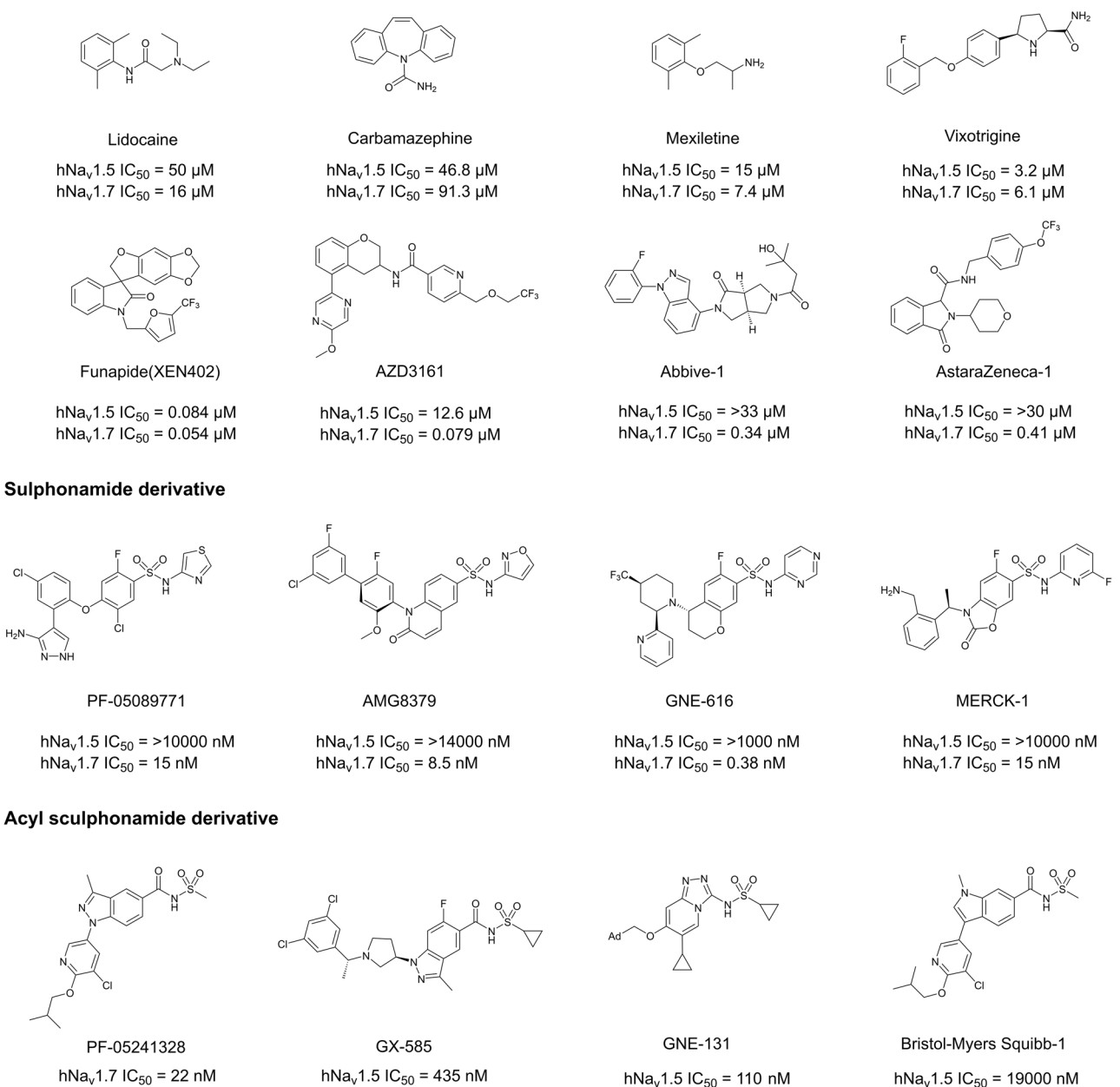

**Fig. 1 | Current development of Nav1.7 inhibitors.** Three types of inhibitors have been developed including: convetional VGSC inhibitors, sulphonamide derivatives and acyl sulphonamide derivatives.

the strategy of V-SYNTHES screening to run a scaffold-oriented screening. During the screening, a library of scaffolds was screened, and the favored ones were selected to generate sub-library by adding different substituents. Then the second-round screening was performed on the sub-library to give in silico hits (Source data of *virtual* screening can be found in supplementary data 10.).

The CIRs which fully use the amphiphilicity of carbenoids are characterized with high efficiency of molecule enumerating, fast reaction rate, good reaction selectivities, and minimal waste production (Fig. 2)[30–36]. Therefore, the CIRs are suitable for the construction of REAL libraries and for discovering pharmaceutically interesting compounds. To save computation cost, we intended to screen for a privileged scaffold for a certain target to develop REAL library. Core skeletons rapidly enumerated based on 150 CIRs were screened via docking to Nav1.7 channel which is a promising target for chronic pain[37,38]. Oxindole scaffold is the most

frequently presented in the preliminary hits (Fig. 2). In addition, the oxindole scaffold can be easily introduced by every substrate of the CIRs, facilitating the growth of an oxindole-based readily accessible library (OREAL) (Fig. 2). Lastly but not least, oxindole-containing compounds have been approved in the clinic use such as Semanib, Sunitinib, and Torceranib, indicating its advantages in the drug-like properties. Thereby, oxindole was chosen as the privileged scaffold for the following library development.

We adopted 22 reactions CIRs to construct an OREAL and gain 2 hits after two round screening. Then we conducted Nav1.7 inhibitory activity assessment, molecular simulations and Paclitaxel-induced neuropathic pain (PINP) model on **C4** and found C4 has a good reversal effect on PINP. Here we show the construction and characterization of the readily accessible OREAL, facilitating the discovery of Nav1.7 inhibitors as a promising therapeutic for PINP.

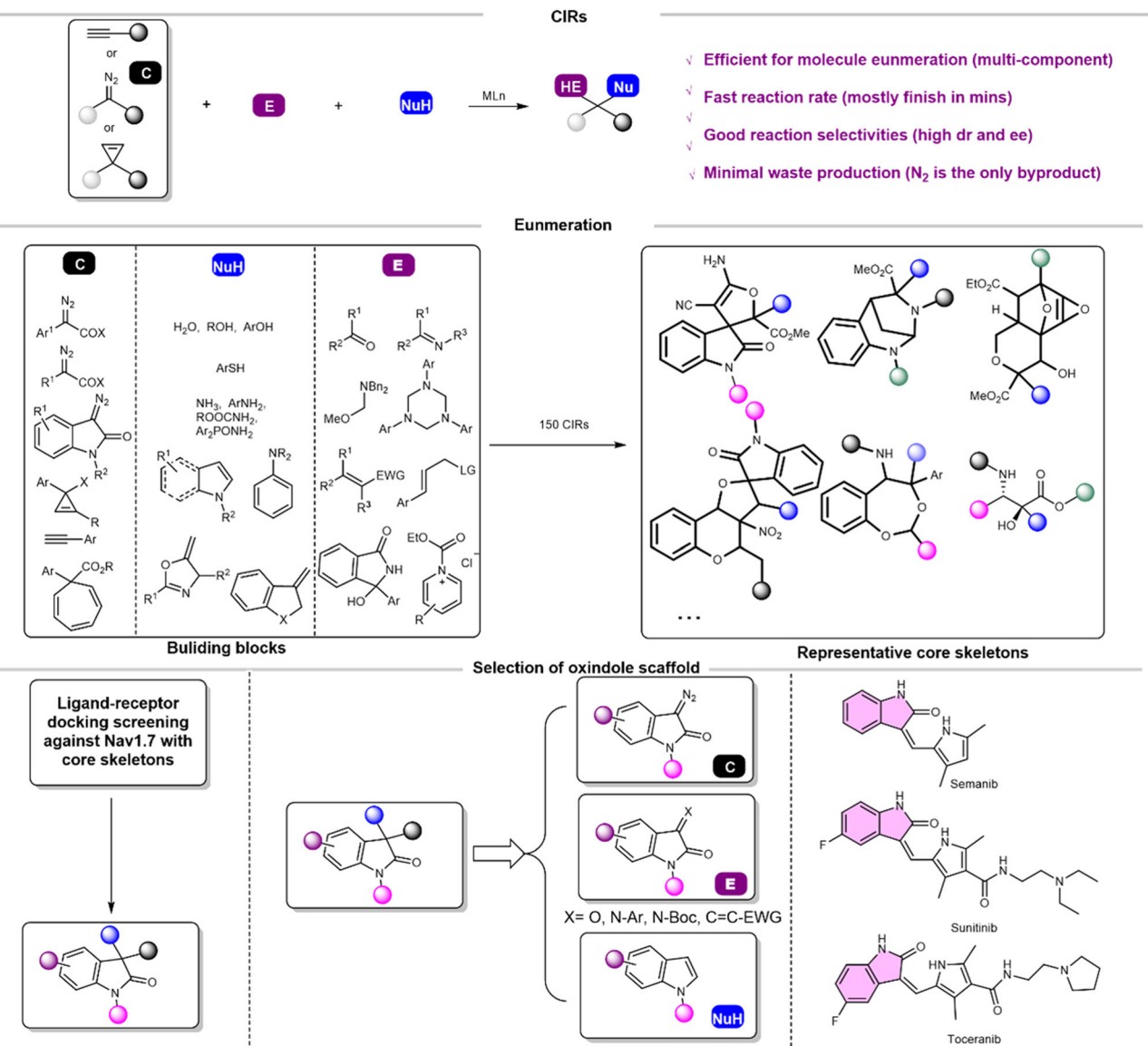

**Fig. 2 | Development of the OREAL based on the CIRs.** Top: the advantages of CIRs for building libraries; Middle: the representative starting materials and products of CIRs; Bottom: The rational of selecting oxindole scaffold.

## Result
### The total step of the construction of OREAL and virtual screening of Nav1.7 inhibitor

We adopted 22 reactions CIRs to construct an OREAL by considering the substrate scope of the CIRs (Fig. 3, step 1). To ensure the synthetic accessibility, we summarized the range of substituent adaptation for each reaction through manual screening. Then we form a smaller scaffold library base on Bemis-Murcko structure by using KNIME (Fig. 3, step 2). The resulting scaffold library was docked to Nav1.7, affording scaffolds with Gibbs energy advantage in binding to the target (Fig. 3, step 3) by using MOE (Molecular Operating Environment). Then we extracted the sub-library from the entire OREAL based on first round screening by using KNIME. The compounds in the sub-library shared same Bemis-Murcko structure with the 18 scaffolds which passed the first round screening. In second round of docking to Nav1.7, we gained 42 virutal hits (Fig. 3, step 4 and 5). The hits were rapidly synthesized via the CIRs to continue bioactivity validation (Fig. 3, step 6 and 7) and resulted in 2 compounds inhibiting the total $Na^+$ ion channels. Subsequently, further Nav1.7 inhibitory activity assessment and molecular simulations were conducted on the most potent compound **C4** (Fig. 3, step 8). Lastly, the compound **C4** was tested on the Paclitaxel-induced

neuropathic pain (PINP) model and effectively reversed PINP. In each step of the operation, we use publicly available cheminformatics software and commercial molecular docking software.

### OREAL holds potential to discover pharmaceutically interesting molecules

To begin with, we generated in virtual molecules by using open-source cheminformatic tool RDKit[39] (https://www.rdkit.org/) based on the CIRs. To ensure the synthetic accessibility of the molecules, the substrate scope of the reactions was considered by following the substrate scope description in our reported methodologies. As shown in Fig. 4A, the isatin and its derivatives involved CIRs were chosen as reactions to construct an OREAL. All the reactions collected were synthetic methodologies published by our group since 2015. In addition, the reactions for synthetic applications of methodologies were also included (specific reactions and corresponding articles are shown in Supplementary Table 1).

The reactants used for generation were first filtered based on the corresponding literature of model reactions to ensure the virtual products can be synthesized in lab. In this work, we present a generation of one reaction's virtual compound library as an example (Fig. 4B). To ensure the synthetic

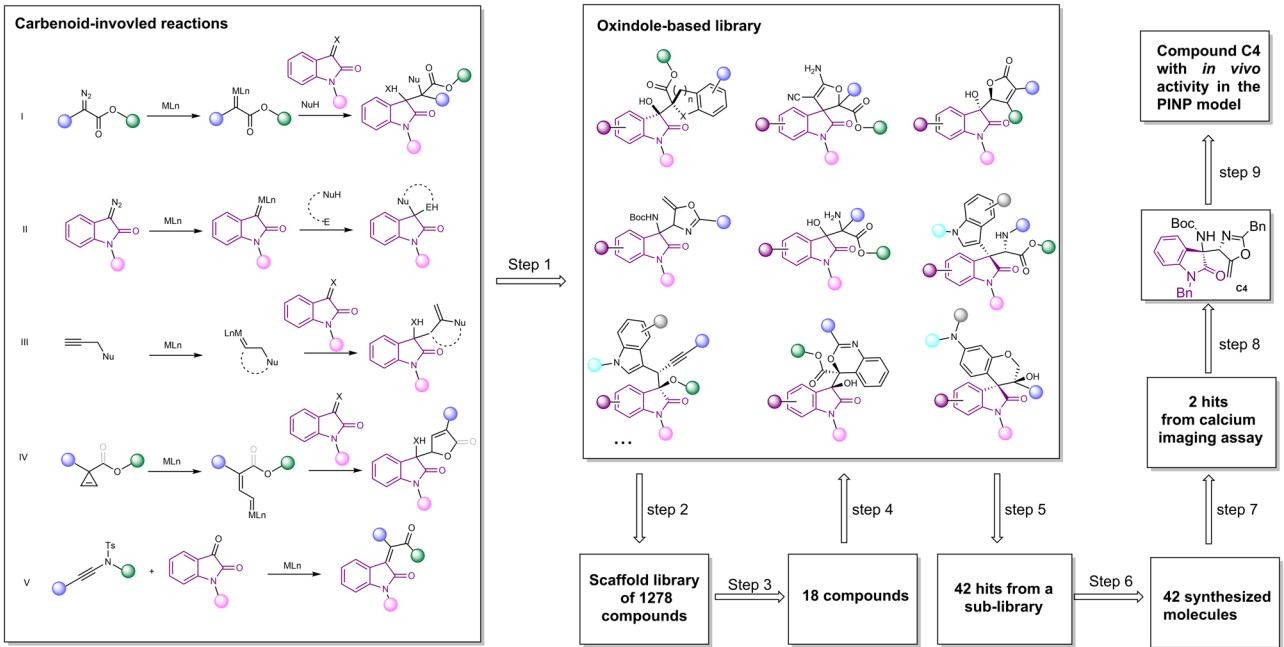

**Fig. 3 | Workflow of discovery of compound C4 as a Nav1.7 inhibitor.** Step 1: Substrate scope directed construction of OREAL via rdkit and OREAL characterization; Step 2: Extraction of scaffolds; Step 3: Molecular docking to Nav 1.7; Step 4: Back to OREAL to search for molecules with desired scaffolds; Step 5: Molecular docking to Nav 1.7 again; Step 6: Organic synthesis of hits; Step 7: Calcium imaging assay screening; Step 8: Nav1.7 inhibitory activity assessment and molecular simulations; Step 9: Investigation of in vivo activity on the PINP mice model.

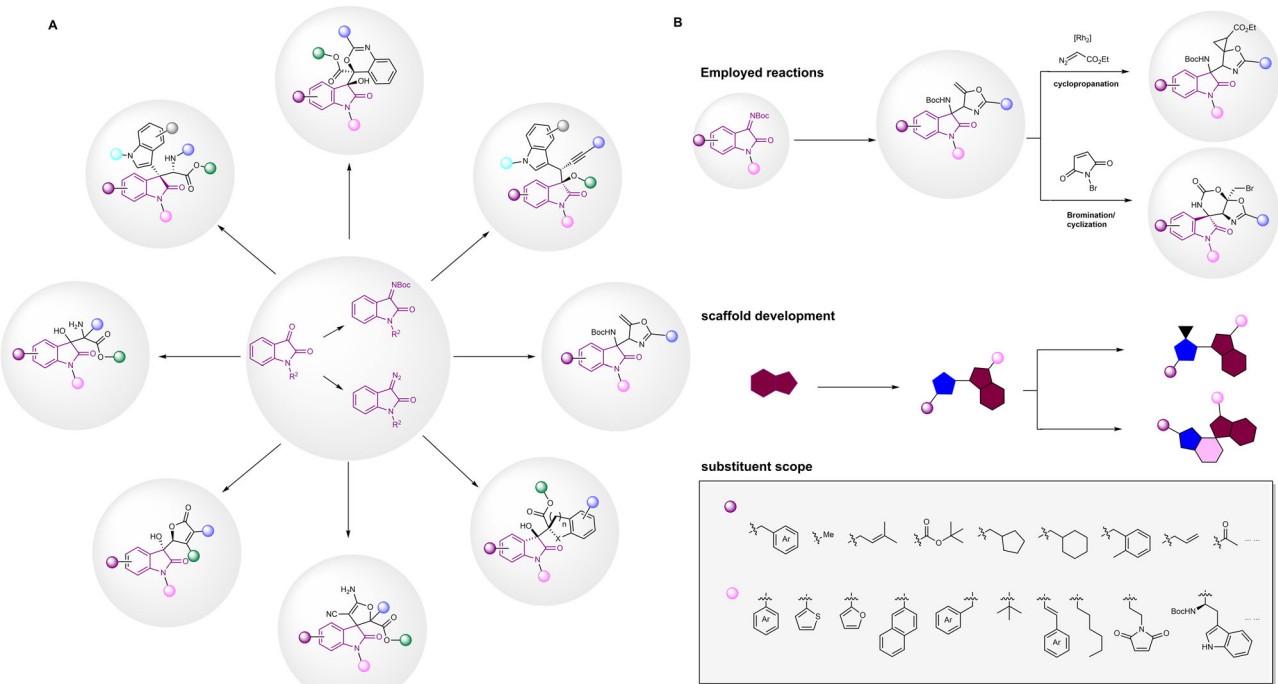

**Fig. 4 | Construction and characterization of OREAL. A** CIRs of 22 papers were employed to build the OREAL. **B** Example of substituents of compound and its derivatives. All derivations shown were from one paper. The derivable sites have been pointed out and available substituents were reported in paper. All collected works are shown in Supplementary Table 1.

accessibility of virtual compounds, we screened the substituents of reactants and determined the characteristics of substituents that each reactant can contain. Due to the large vast scale of molecules in the OREAL, we have adopted the Bemis-Murcko structure to classify the OREAL library[40]. Bemis-Murcko structure is used for analyzing the shape and structure of drugs. This method decomposes molecules into four types of units: ring system, linker, scaffold, and side chain. Notably, scaffold which is composed of a ring system and a linker, and side chain are combined to form drug molecules. It should be noted that in addition to the core structure, substituents are play important roles in the bioactivities as well[41,42]. In the light of this, we continued to expand the compound library by adding the substituents by following the reported substrate scope of the reactions and

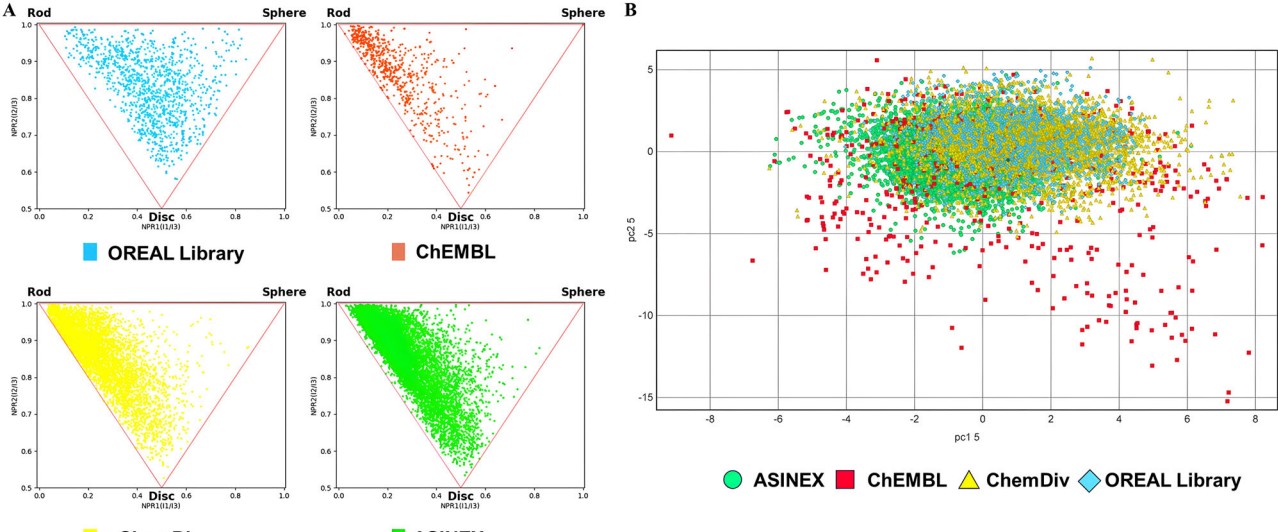

**Fig. 5 | Characterization of OREAL and commercial library. A** Principal Moment of Inertia (PMI) analysis of four libraries: OREAL (Blue), ChEMBL (Red), ChemDiv (yellow) and ASINEX (green). **B** PCA analysis of different compound libraries. Different libraries have been shown in different colors: OREAL (Blue), ChEMBL (Red), ChemDiv (yellow) and ASINEX (green). The loadings of PC1 and PC2 are listed in Supplementary Fig. 6. 11 descriptors were used for PCA, including hydrogen bond donors (HBD), hydrogen bond acceptors (HBA), number of rotatable bonds (RB), molecular weight (MW), octanol/water partition coefficient (logP(o/w)), and topological polar surface area (TPSA), number of heavy atom (NH), number of chiral center (chiral), fraction of rotatable bond (FRB), number of ring(ring) and number of aromatic atom (NA). Each library's PCA analysis is shown separately in four independent figures in Supplementary Fig. 7.

finished building of whole OREAL. The multiple substitutable sites in the substrates of CIRs allow us to rapidly expand the scaffold library into a much bigger library. Compared with mono- or bi-component reactions, multi-component reactions provide more structurally diverse molecules in a single step[43–45]. This advantage benefits the diversity of compound libraries and increases the hit rate. Among the employed reactions, many multi-component reactions[35,46–48] play a crucial role in quick expansion of the OREAL. In general, over 20 million virtual molecules were generated with 1278 scaffolds by using 39 reactions in 22 synthetic methodology papers.

To characterize the quality of the OREAL, we conducted a comparison between our library and commercially available libraries to analyze the overall composition of the OREAL library. We used scaffold library for comparison. The chemical space of the OREAL library was compared with 3 different commercial compound libraries, compounds with sodium channel inhibitory activity included in ChEMBL[49–51], AN4500 Ion Channel library from ASINEX and Voltage-Gated Ion Channel-Targeted Library from ChemDiv(www.chemdiv.com), by using Principal Moment of Inertia (PMI)[52] analysis and Principal Component Analysis (PCA)[53,54] To explore the characteristics of shapes of virtual compounds, a PMI analysis was taken to visualize the 3D shape diversity of our OREAL. The principal component analysis was performed by RDKit's rdMolDescriptors to calculate three PMI values (I1, I2, and I3) for each conformation and further calculated npr1 (I1/I3) and npr2 (I2/I3) for PMI plot. As shown in Fig. 5A, the PMI analysis revealed that the OREAL has a unique 3D shape compared to other libraries. The compounds in the OREAL are evenly distributed in shape and lean towards spherical shapes. This indicates that the 3D shape of molecules in OREAL library is unique, leaving a chance of searching for hits with novel structures.

PCA analysis is a dimensionality reduction algorithm that can reduce multiple variables on a two-dimensional plane for visual analysis. Through PCA analysis, the chemical space of OREAL and other compound libraries with sodium ion channel inhibitory activity can be visualized. By calculating the various chemical properties of compounds in different compound libraries, which are often related to compound's druggability and drug-likeness, and analyzing them through dimensionality reduction algorithms, the chemical space is drawn based on molecular properties, and finally chemical spaces of different compound libraries are compared.

To compare the chemical space of the OREAL with that of other commercial libraries, we adopted Principal Component Analysis (PCA) using Datawarrior[55]. The PCA analysis revealed that the relationship between Principal Component 1 (PC1) and Principal Component 2 (PC2) accounts for 57.4% of PCA data variability. The detailed contribution of each calculated value is presented in Supplementary Figs. 6 and 9. Our OREAL has a similar chemical space with reported sodium channel inhibitor compounds from TargetMol and ChemDiv, while the compounds in ChEMBL library are different from the other three libraries. Figure 5B shows that the compounds in the OREAL share similar drug-like properties with the other commercial libraries, ensuring an ideal hit rate for the upcoming screening.

### Virtual screening of Nav1.7 inhibitors generates 42 hits
The allosteric modulation of Nav1.7 is vital to gain isoform-selectivity which is essential to avoid side effects. Thus, the VSD IV domain, a known allosteric site of Nav1.7, has been chosen to perform the upcoming docking screening. Targeting the VSD IV binding domain of the Nav1.7 target has gained much attention in the academic community[56–58]. According to the literature, Tyr1537, Trp1538, Arg1602, and Arg1608 residues are actively involved in the binding events of Nav1.7 inhibitors. The process of virtual screening of the OREAL was shown in Fig. 3. The entire screening consists of two main stages. In the first stage, 1278 scaffolds from OREAL were screened to find the privileged scaffolds that match the allosteric binding site of Nav1.7. 14 scaffolds have passed through the first-round screening. Then, the molecules sharing the same Bemis-Murcko structure with the selected scaffolds were extracted from the OREAL and used in the second-round docking screening (for each scaffold, there will be 300-1000 compounds, which were monosubstituted from corresponding scaffold, used for the next screening). Therefore, the interactions with the four residues were evaluated during visual inspection. Resultingly, 42 virtual hits were generated and synthesized for the following bioactivity validation (Supplementary Table 3).

### C4 is the best candidate in the inhibition of VTD-evoked neuronal activities
Veratridine (VTD) is a voltage gated sodium channel (VGSC) regulator used as an agonist in the functional screening of VGSC blockers. By using

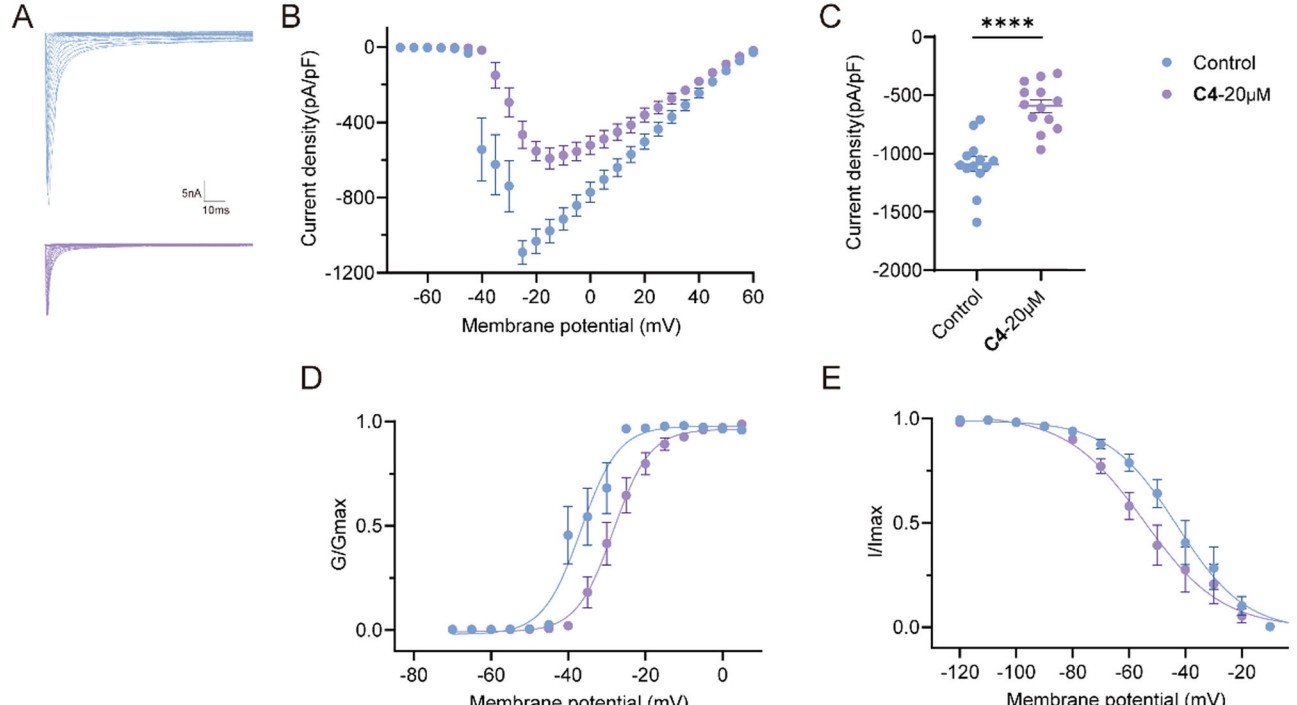

**Fig. 6 | C4 inhibited total Na⁺ currents in rat DRG neurons. A** Representative traces of Na+ currents from DRG sensory neurons treated with 0.1% DMSO (control) or 20 μM **C4**. Currents were evoked by 150 ms pulses between −70 and +60 mV. **B** Summary of the normalized (pA/pF) sodium current density versus voltage relationship. **C** Normalized peak currents from DRG sensory neurons treated as indicated (****$P < 0.0001$, unpaired two-tailed Student's t test, n > 12 cells per condition). Boltzmann fits for normalized conductance voltage relations for voltage dependent activation (**D**) and inactivation (**E**) of sensory neurons treated as indicated. Asterisks indicate statistical significance compared with cells treated with 0.1% DMSO. Values for V1/2, the voltage of half-maximal inactivation and activation, and the slope factors (k) were derived from Boltzmann distribution fits to the individual recordings and were averaged to determine the mean (±SEM) voltage dependence of steady-state inactivation and activation, respectively. Source data of this figure can be found in supplementary data 1.

the VTD-evoked calcium imaging assay, the inhibition rate of compounds on sodium channels can be indirectly measured. To validate the inhibitory activity of the 42 virtual hits, firstly, we performed the VTD-evoked calcium imaging assay as previously described[59,60]. VTD activates non-selective Nav channels, resulting in channel opening and influx of sodium ions[61]. Out of the 42 compounds, **2** compounds showed inhibitory activity against Na⁺ ion channels (cutting threshold: inhibition rate >50%), with compound **C4** exhibiting the highest inhibition on VTD-evoked neuronal responses by reducing the response (ΔF/F0) to around 66.7% (For details see Supplementary Fig. 8, source data of the figure can be read in supplementary data 5).

### C4 Inhibited Total Na⁺ Currents in Rat DRG

To test the inhibitory potential of **C4** on total sodium currents, whole cell voltage-clamp experiments were carried out on small-diameter rat DRG sensory neurons. Representative traces of total Na⁺ currents recorded from DRG neurons treated with 0.1% DMSO or 20 μM **C4** were displayed in Fig. 6A. Approximately 50% of the sodium peak currents density were inhibited by **C4** (Figs. 6B, C; 0.1% DMSO, −1090.8 ± 60.5 pA/pF; **C4**, −592.8 ± 55.5 pA/pF). Since this current reduction could result from channel gating adjustments, we investigated whether **C4** could alter the voltage-dependent activation and inactivation properties of sodium currents in DRG neurons (Figs. 6D, E). Comparing the midpoints ($V_{1/2}$) and slope factors (k) in response to changes in command voltages of whole-cell ionic conductance enabled the measurement of the activation and inactivation changes of sodium currents for DRG cells treated with the compounds. There was a hyperpolarizing shift of ~8 mV in the $V_{1/2}$ of steady-state inactivation and a depolarizing shift of ~12 mV in the $V_{1/2}$ of activation in the presence of **C4** (Supplementary Table 4), while both the slopes were unaffected. The result indicated that **C4** accelerates the inactivation of sodium channels while decelerating their activation. To account for

population diversity in terms of neuronal size, these peaks were normalized by cell capacitance and subsequently displayed as peak current density (pA/pF).

### C4 is more potent at blocking tetrodotoxin-resistant Nav Channels than tetrodotoxin-sensitive Nav channels of DRG Neurons

Nav channels are characterized into tetrodotoxin-resistant (TTX-R) and tetrodotoxin-sensitive (TTX-S) Nav channels in terms of their sensitivity to tetrodotoxin (TTX)[62]. To further understand the interaction of **C4** on voltage-gated Na⁺ channels, we dissected TTX-resistant sensitive sodium currents by 1 μM TTX and TTX-sensitive sodium currents by 1 μM A803467[63] (Nav1.9 currents was inactivated by holding the neuron at -50mV for 500 ms as previously reported[64]). The results showed that the IC50 for TTX-S Nav channels was approximately 40 times lower than that for TTX-R Nav channels (0.44 μM vs 18.29 μM, Figs. 7J, D). The studies suggested that **C4** was more potent at blocking TTX-S than TTX-R Nav channels in a neuropathic pain state, particularly in inactivated channels.

### C4 selectively targeted Nav1.7 to inhibit Na⁺ currents

Since Nav1.7 carries a large portion of the TTX-S Na⁺ current in small diameter DRG neurons and **C4** has more potent at blocking TTX-S Na⁺ current, we asked whether **C4** inhibited the Na⁺ currents mainly through Nav1.7 channel. To investigate our hypothesis, we utilized HEK293 cell lines transfected with pcDNA3.1-Nav1.7-Flag and GFP plasmid, which were then incubated overnight with either 0.1% DMSO or multi-concentration **C4** for whole cell patch recording. The greater concentration of **C4** led to a more significant inhibition of the current density, indicating a dependency on concentration (Figs. 8B, C, 0.1% DMSO, −701.2 ± 93.1 pA/pF; 0.1 μM **C4**, -536.6 ± 107.8 pA/pF; 1.0 μM **C4**, −501.9 ± 137.7 pA/pF; 2.0 μM **C4**, −349.8 ± 91.8 pA/pF; 5.0 μM **C4**, −108.7 ± 20.2 pA/pF; 10.0 μM **C4**, -80.1 ± 12.2 pA/pF; 15.0 μM **C4**, −89.3 ± 16.7 pA/pF, temporary effect of

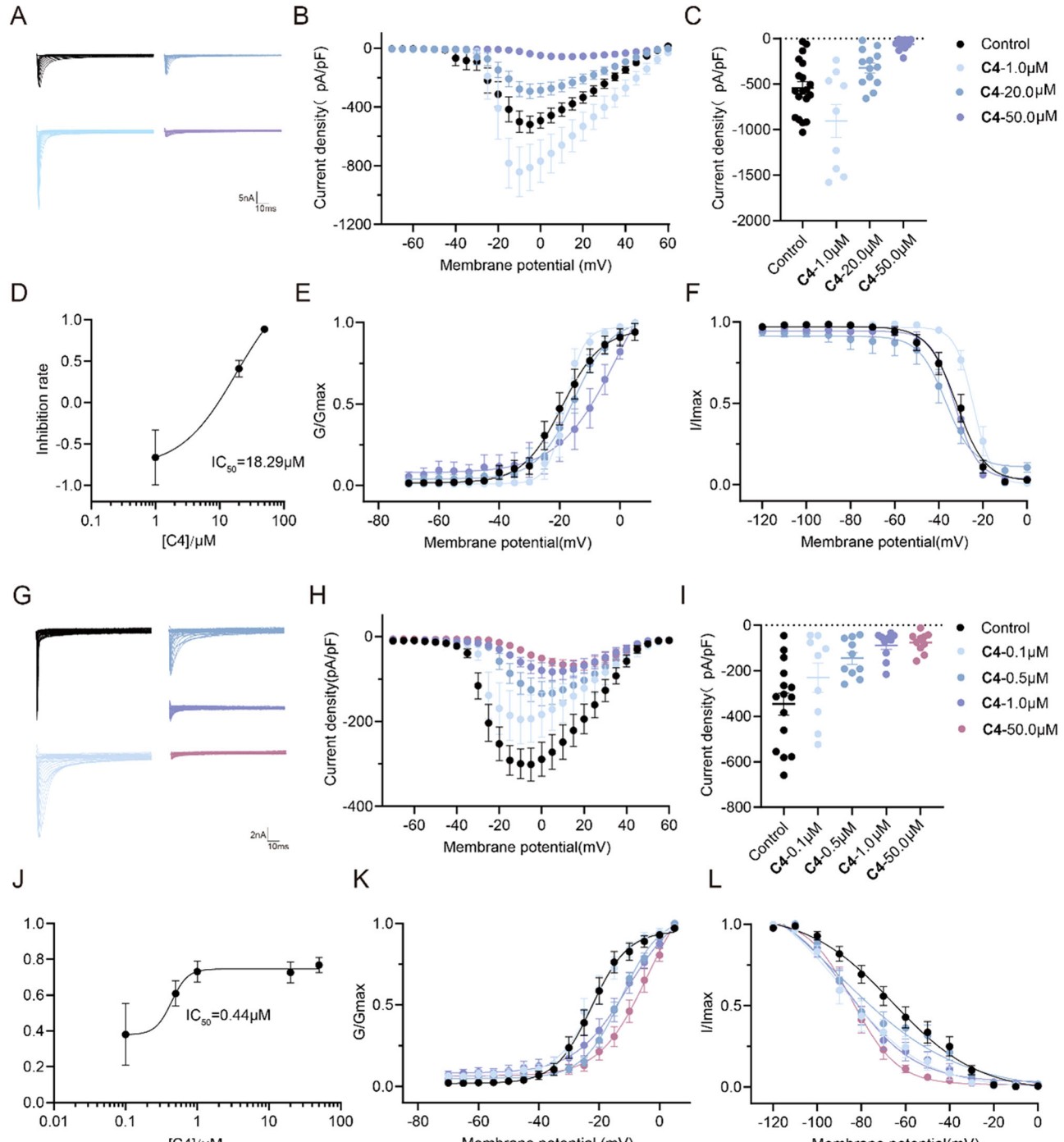

**Fig. 7 | The effect of C4 on TTX-R channels and TTX-S channels in DRG neurons.**
**A** Representative traces of TTX-R Na+ currents from DRG sensory neurons treated with 0.1% DMSO (Control) or 1 μM, 20 μM or 50 μM **C4**. **B** Summary of the normalized (pA/pF) TTX-R sodium current density versus voltage relationship. **C** Normalized peak currents from DRG sensory neurons treated as indicated (n > 8 cells per condition, one-way ANOVA test, 1 μM **C4,** P = 0.8335; 20 μM **C4,** P = 0.3122, 50 μM **C4,** ****P < 0.0001). **D** Concentration curve of inhibition rate by **C4** on TTX-R Na+ currents in DRG neurons. Boltzmann fits for normalized conductance voltage relations for voltage dependent activation (**E**) and inactivation (**F**) of sensory neurons treated as indicated. Asterisks indicate statistical significance compared with cells treated with 0.1%

DMSO. **G** Representative traces of TTX-S Na+ currents from DRG sensory neurons treated with 0.1% DMSO (Control) or 0.1 μM, 0.5 μM, 1 μM or 50 μM **C4**. **H** Summary of the normalized (pA/pF) TTX-R sodium current density versus voltage relationship. **I** Normalized peak currents from DRG sensory neurons treated as indicated (n > 8 cells per condition, one-way ANOVA test, 0.1 μM **C4,** P = 0.3917; 0.5 μM **C4,** P = 0.0632;1 μM **C4,** ***P = 0.0008, 50 μM **C4,** ***P = 0.0005). **J** Concentration curve of inhibition rate by **C4** on TTX-S Na+ currents. Boltzmann fits for normalized conductance voltage relations for voltage dependent activation (**K**) and inactivation (**L**) of sensory neurons treated as indicated. Source data of this figure can be found in supplementary data 2.

20 μM **C4** on the Nav1.7-transfected HEK293 cells can be found in Supplementary Fig. 10, source data can be read in supplementary data 7). The results showed that the $IC_{50}$ value for Nav1.7 channels was around 2.23 μM, which was in line with the $IC_{50}$ value for TTX-S Nav channels. The $V_{1/2}$

value of activation and inactivation also changed with the concentration of **C4** raised (Fig. 8E, F and Supplementary Table 4), while the slope of activation and inactivation remained constant. We also explored the activity of **C4** in other TTX-S Nav channels (Supplementary Table 5, source data can

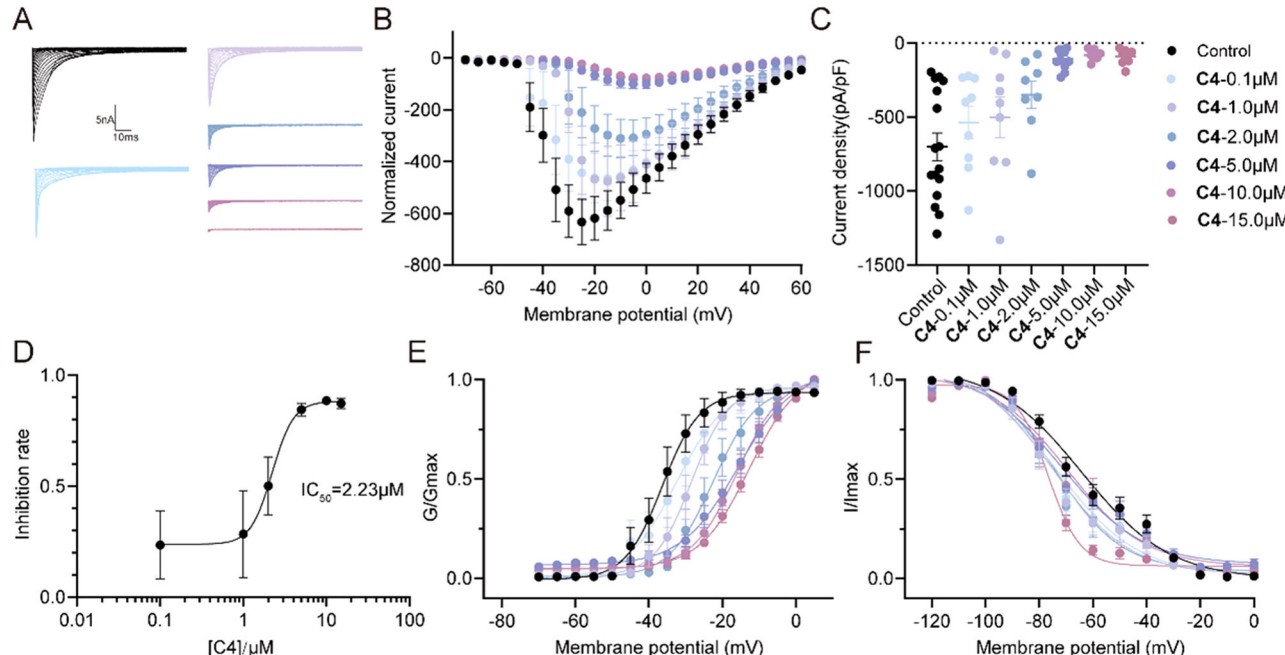

**Fig. 8 | C4 selectively targeted Nav1.7 to inhibit Na+ currents. A** Representative traces of Nav1.7 currents from Nav1.7-transfected HEK293 cell line treated with 0.1% DMSO (Control) or 0.1 μM, 1.0 μM, 2.0 μM, 5.0 μM, 10.0 μM or 15.0 μM **C4**. **B** Summary of the normalized (pA/pF) Nav1.7 current density versus voltage relationship. **C** Normalized peak currents from Nav1.7-transfected HEK293 cell line treated as indicated ($n > 7$ cells per condition, one-way ANOVA test, 0.1 μM **C4**, $P = 0.8128$; 1.0 μM **C4**, $P = 0.7864$; 2.0 μM **C4**, $P = 0.0799$; 5.0 μM **C4**, ****$P < 0.0001$; 10.0 μM **C4**, ****$P < 0.0001$; 15.0 μM **C4**, ****$P < 0.0001$). **D** Concentration curve of inhibition rate by **C4** on Nav1.7 in Nav1.7-transfected HEK293 cell line. Boltzmann fits for normalized conductance voltage relations for voltage dependent activation (**E**) and inactivation (**F**) of Nav1.7-transfected HEK293 cell line treated as indicated. Source data of this figure can be found in supplementary data 3.

**Fig. 9 | C4 reversed PINP. A** The 50% paw withdraw threshold of adult male rat ($n = 6$) were measured at 21 days post PTX-injected. **B** Area under curve for mechanical pain behavior. Data are presented as means ± SEM. Asterisks indicate statistical significance compared with vehicle treatment (PTX, 17.7 ± 2.7; **C4**, 37.95 ± 5.3; **$P < 0.01$ and ****$P < 0.0001$; two-way ANOVA with Sidak's post hoc test). The experimenter was blinded to the treatment condition. Source data of this figure can be found in supplementary data 4.

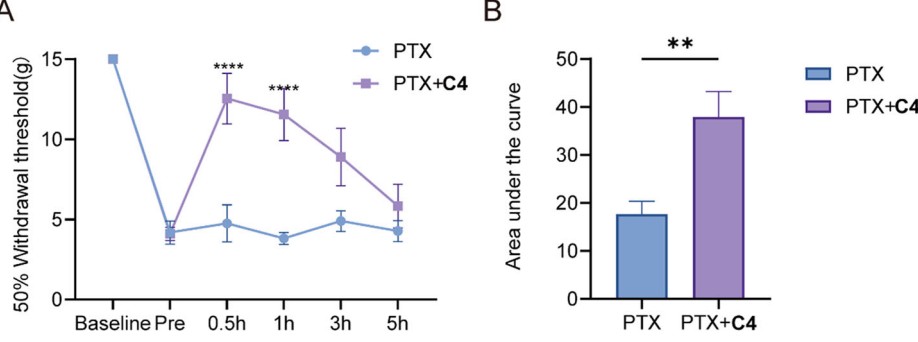

be read in supplementary data 8). These findings indicated that **C4** selectively targeted Nav1.7 to inhibit Na$^+$ currents.

## C4 effectively reverses PINP

To evaluate the potential of **C4** to reverse nociception, we established a PINP model. Multiple intraperitoneal injections (days 0, 2, 4, and 6) of PTX (2 mg/kg/d, cumulative dose of 8 mg/kg) produced lasting mechanical allodynia at 3 weeks measured by Von-Frey test. Intrathecal injection of **C4** significantly increased paw withdrawal thresholds (PWTs). The anti-nociception effect of **C4** peaked at 0.5-hour and persisted for nearly 5 hours (Fig. 9A). Area under the curve also showed a significant antinociceptive effect of **C4**-treated animals compared to vehicle-treated animals (Fig. 9B). The result indicated that **C4** is antinociceptive in rodent models of neuropathic pain.

## C4 is not a cardiac liability

To investigate potential cardiac toxicity, we evaluated the impact of 20 μM **C4** (a high concentration) on hERG (human Ether-à-go-go-Related Gene) current transient expressed in HEK293 cells. After control currents were

recorded, the cell was recorded with the same protocol during 10 min perfusion of 20 μM **C4**, followed by a 10 min washout. Supplementary Fig. 10 shows that **C4** caused only a weak change of hERG current, with an inhibition of 0.97% ± 2.9%, no significant difference. The results indicated that **C4** has no effect on cardiac function (Supplementary Fig. 9, source data can be read in supplementary data 6).

## The molecular docking and molecular dynamic analysis of C4 suggest the binding mode of C4-5EK0 complex

Considering the **C4** possible mechanism with 5EK0, we intended to gain deep understanding on the interaction mode between **C4** and Nav1.7. To begin with, a molecular docking experiment was conducted between **C4** and Nav1.7. Consequently, the binding moieties of **C4** could be mainly divided into two parts. The indole ring had a π - π stacking effect with Tyr1537 and Trp1538, while the benzyl group extended into the cavity and had a pi-π stacking effect with Arg1602 (Fig. 10). To ensure the accuracy of the docking model, a molecular dynamic analysis was taken for further study and MM/PBSA approach, and indicates that the docking model is stable, with the binding free energy of -129.204 kJ/mol (Supplementary Table 2, source data

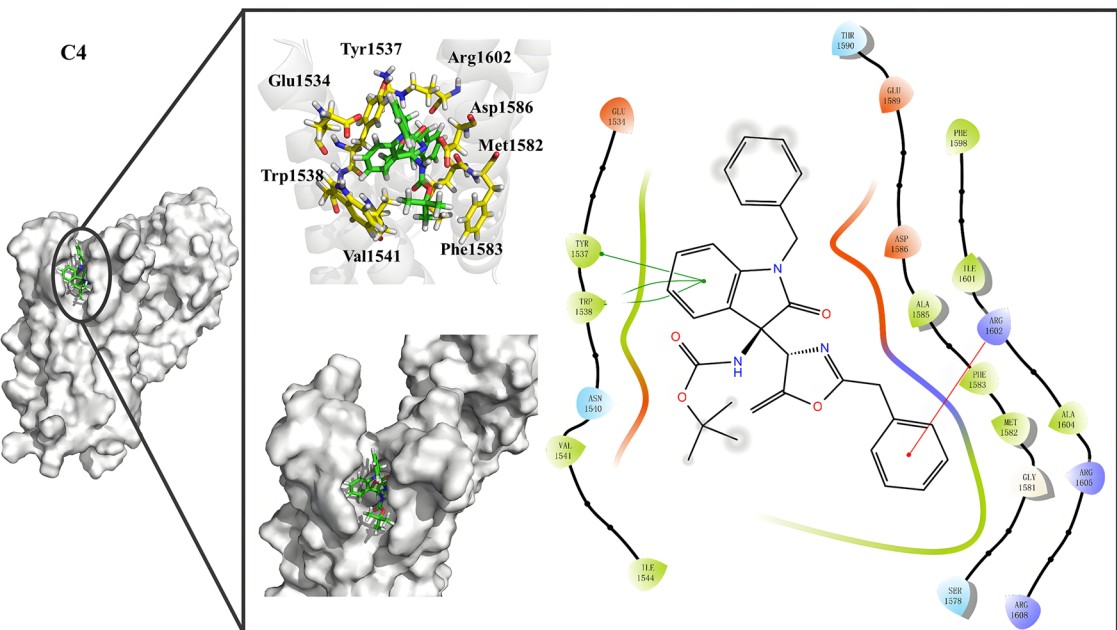

**Fig. 10 | Binding mode of ligand C4.** The binding model of C4-5EK0 complex shows in cartoon model, surface model and 2D interaction picture. The ligand C4 mainly have interaction with Tyr1537, Trp1538 and Arg1602 in magnified docking pose.

of the MD simulation can be read in supplementary data 9). The MD simulation result was displayed in Supplementary Section 3 and Supplementary Fig. 1, 2, 3, 4, 5.

## Discussion

The drug discovery campaign has been significantly influenced by the fast-evolving technologies of computation. While the bottleneck of computation-based screening is difficult access to virtual hits, thus delaying the validation of pharmaceutical activities. The REAL library screening and V-SYNTHES screening have opened a wide window for searching for interesting drug candidates from the ultra-large library of $10^{60}$ drug-like compounds. The full disclosure of the interactions between drug-like compounds and targets requires the co-work of synthetic methodologies and cutting-edge screening technologies. To this end, we used a scaffold-based screening strategy to build and screen the OREAL, which is based on the CIRs developed by our group in the last decade. The OREAL is otherwise inaccessible and has great potential to discover pharmaceutically interesting molecules. Indeed, compound **C4** with a unique structure is screened out as a potent Nav1.7 inhibitor by docking to the VSD IV domain of Nav1.7. A careful characterization of the bioactivity of **C4** revealed that **C4** is a selective Nav1.7 inhibitor and effectively reverses PINP in rodent models. Preliminary toxicology studies showed that **C4** is negative to hERG, implying that **C4** has no effect on cardiac function. Consistent results of molecular docking and molecular simulations further support the reasonability of the in-silico screening and show the insight of the binding mode of C4. Nav1.7 as a target for the treatment of chronic pain has drawn much attention in the drug discovery. However, few potent Nav1.7 inhibitors show efficacy in preclinical pain models or human pain models[65]. The unexpected results are mainly attributed to the inappropriate pain model used. For instance, cellularly potent Nav1.7 inhibitors are evaluated on the acute pain model, instead of the chronic pain models that simulate the clinical conditions. We tested the effect of compound **C4** on the PINP model, which is closely related to the clinical conditions of chemotherapy-induced peripheral neuropathic pain (CIPN)[66]. Compound **C4** represents one of the few Nav1.7 inhibitors that show a significant effect on the PINP model, rendering a chance to develop non-opioid analgesic drugs for PINP. In the subsequent work, we will further optimize the structure of C4 to optimize its existing drug activity by using a more accurate human Nav1.7 crystal model[67]. Overall, we constructed the OREAL and generated a selective Nav1.7

inhibitor with the potential to treat CIPN, paving the way for pushing Nav1.7-based drug candidates forward to the clinic.

## Methods
### Generation of OREAL
The OREAL is built by using cheminformatics tools RDKit[39] (https://www.rdkit.org/). After confirming the collected reaction, we convert reaction into SMARTS language for further generation of the library. Some multi-component reactions were divided into several steps based on their mechanism to simplify the compound generation protocols. We generate the virtual scaffold library base on Bemis-Murcko structure[40]. After the construction of virtual scaffold library, we expand the compound library by adding the substituents on each structure based on reaction boundary of each reaction and finish building of whole OREAL. Before second screening, we used KNIME(version 4.5.3, Released Day: February 1, 2022) to search the compounds which shared same Bemis-Murcko with first round screening compounds.

### Chemical space exploration
PMI analysis was taken to visualize the 3D shape diversity of the OREAL. Firstly, the 3D conformation of each virtual compound was generated by Ligprep protocol in Schrödinger software (Release 2019-2, Schrödinger LLC, New York, NY, 2019). The principal component analysis was taken by RDKit's rdMolDescriptors to calculate three PMI values (I1, I2, and I3) for each conformation and further calculate npr1 (I1/I3) and npr2 (I2/I3) for PMI plot.

The Principal Component Analysis was taken by using Datawarrior[55]. 11 descriptors were used for PCA, including hydrogen bond donors (HBD), hydrogen bond acceptors (HBA), number of rotatable bonds (RB), molecular weight (MW), octanol/water partition coefficient (logP(o/w)), and topological polar surface area (TPSA), number of heavy atom (NH), number of chiral center (chiral), fraction of rotatable bond (FRB), number of ring(ring) and number of aromatic atom (NA). After the descriptors data was collected and calculated by MOE, the PCA was taken by using Calculate Principal Component function in Datawarrior.

### Virtual Screening
The process of using MOE (The Molecular Operating Environment, Chemical Computing Group Inc. Version 2019.0102) for virtual screening is as

follows. The protein structure (PDB: 5EK0)[68] was added hydrogen atoms and minimized energy by using the QuickPrep protocol (the water molecules farther than 4.5 A were deleted and RMS gradient was fell below 0.1 kcal mol-1 Å-1 under MMFF94X force field). The binding site was defined same as the GX-936 site and the docking method was Traiangle Matcher method, scoring with London ΔG scoring function. The refinement scoring function was GBVI/WSA ΔG Method. Each ligand was calculated placement for 30 poses. The highest scoring poses for each ligand remained. It should be noted that during two round screenings, ligands with interaction patterns with Tyr1537, Trp1538, Arg1602, and Arg1608 residues were prioritized.

## MD simulation and calculation of binding free energies

MD simulation adopts Gromacs 2022.1 program[69] under constant temperature and pressure and Periodic boundary conditions. Protein application Amber14SB force field[70], small molecule application based on Amber's GAFF force field, TIP3P water model[71]. In the MD simulation process, all bonds involving hydrogen atoms are constrained using the LINCS algorithm[72], with an integration step of 2 fs. The electrostatic interaction is calculated using the Particle mesh Ewald (PME) method[73]. The non bond interaction truncation value is set to 10 Å and updated every 10 steps. The V-rescale[74] temperature coupling method was used to control the simulated temperature to 298.15 K, and the Parrinello Rahman method[75] was used to control the pressure to 1 bar. Firstly, the steepest descent method is used to minimize the energy of the two systems to eliminate close contact between atoms; Then, perform NVT and NPT equilibrium simulations at 500 p es under 298.15 K conditions; Finally, a 100 ns MD simulation was performed on the system, with conformations saved every 10 ps. The visualization of the simulation results was completed using the Gromacs embedded program and VMD.

## Animal study

Adult male Sprague-Dawley rats (Beijing Viton Lever) were housed in a pathogen-free facility with controlled temperature conditions ($23 \pm 3$ °C) and a 12-hour light/dark cycle (lights on from 7:00 AM to 7:00 PM). Standard rodent food and water were readily available. All experiments were approved by the Animal Care and Use Committee at Shenzhen University Medical School. All experiments followed regulations for the Care and Use of Laboratory Animals and ethical guidelines from the International Association for the Study of Pain. Behavioral experiments were conducted by experimenter's blind to the experimental groups and treatments.

## Reagents

Unless otherwise specified, the reagents were bought from Sigma-Aldrich Chemicals. (St. Louis, MO, USA). Neutral protease (Worthington, LS02104), IA collagenase (Worthington, LS004194), Poly-D-lysine hydrobromide (Sigma, P7886), DMEM (Gibco, 2318815), Fetal bovine serum (Gibco, 2176404), penicillin/streptomycin (Gibco, 2321127), JetPrime (Polyplus, 101000046), TTX (Alomone, T-550), A803467 (Alomone, A-105).

## Primary dorsal root ganglion (DRG) neuronal cultures

The dorsal root ganglia were acutely dissociated from Sprague-Dawley rats that were euthanized between 4 to 8 weeks of age and anesthetized with isoflurane prior to decapitation. The DRG from all spinal levels were then removed and collected in DMEM at room temperature. Following this, the collected DRG were placed in a mixture of enzymes (3.125 mg/mL neutral protease and 5 mg/mL type IA collagenase, Worthington) for digestion and incubated at 37 °C under gentle agitation for approximately one hour. After being centrifuged at 27 °C and 800 r/min for 3 minutes, dissociated DRG neurons were resuspended in DRG media (DMEM containing 1% penicillin/streptomycin sulfate and 10% fetal bovine serum) and then plated onto glass coverslips previously treated with poly-d-lysine (100 μg/mL). All cultures were used within 24 hours.

## Cell culture and transient transfection of HEK293 cell lines

HEK293 cell lines derived from human embryonic kidney were cultivated in DMEM supplemented with 10% (vol/vol) fetal bovine serum and 1% penicillin/streptomycin sulfate. The cells were maintained in standard conditions consisting of 5% CO2, saturated humidity, and a temperature of 37 °C. The transfection procedure involved using jetPRIME to introduce pcDNA3.1-Nav1.7-Flag or pcmv-kcnh2-egfp-neo plasmid (purchased from Shenzhen Dingke Biotechnology Co., Ltd) with GFP plasmids into cells when confluence levels reached 60-80%. Electrophysiological tests were performed 24-48 hours following cell transfection.

## Veratridine-triggered calcium imaging screen in Acutely Dissociated DRG Neurons

DRG neurons were incubated with 3 μM Fura-4AM (Thermo Fisher, # F14201, stock solution prepared at 1 mM in DMSO) at 37 °C for 30 minutes to investigate changes in intracellular $Ca^{2+}$. Throughout the experiment, the cultured neurons were continuously perfused with a standard bath solution (mM) containing 139 NaCl, 3 KCl, 0.8 $MgCl_2$, 1.8 $CaCl_2$, 10 HEPES, and 5 glucose, pH 7.4 with NaOH. Veratridine (MCE, #HY-N6691, 30 μM) was applied after recording a stable baseline ($F_0$) for 1 minute. The recording lasted for 3 minutes before being switched to a standard bath. At the end of the recordings, Ringers with 90 mM KCl was perfused to distinguish viable neurons. The changes in intracellular Ca2+ concentration was evaluated by calculating the ratio of $\Delta F/F_0$ after subtracting the background. Neurons that responded to 90 mM KCl were included in the analysis. On rare occasions, neurons reacted to VTD but not KCl, or the KCl response was unclear because the $Ca^{2+}$ signal did not return to baseline after the administration of the last agonist. A response is deemed an increase in ($\Delta F/F_0$) ratio of 10% higher than baseline. We conducted $t$-tests to compare mean values obtained from this study. Fluorescence imaging was conducted using an inverted microscope, the Nikon Eclipse Ti-U (Nikon Instruments Inc). The microscope was equipped with a Nikon Plan Fluor 4× 0.13 objective and a Photometrics cooled CCD camera CoolSNAP HQ2, both controlled by NIS Elements software (Nikon instruments, version AR 5.30.03). The excitation light was provided by a current-regulated power supply (HAMAATSU PHOTONICS KK) at 488 nm and was delivered through a Lambda 10-3 system (Sutter Instruments). To minimize photobleaching and phototoxicity, images were captured at 5-second intervals during the experiment. The minimum exposure time was utilized to obtain images of satisfactory quality.

## Whole-cell patch recordings of Na+ currents in acutely dissociated DRG neurons

All recordings were obtained from acutely dissociated DRG neurons from Sprague Dawley rats. For sodium current recordings the internal pipette solution consisted of (in mM): 140 CsF, 10 NaCl, 1.1 Cs-EGTA, and 15 HEPES (pH 7.3, mOsm/L = 290–310) and external solution consisted of (in mM): 140 NaCl, 30 tetraethylammonium chloride, 10 D-glucose, 3 KCl, 1 $CaCl_2$, 0.5 $CdCl_2$, 1 $MgCl_2$, and 10 HEPES (pH 7.3, mOsm/L = 310-315). DRG neurons were recorded with current-voltage (I-V) and activation/inactivation voltage. The voltage protocols for the experiment involved the following steps. For I-V protocol, the cells were patched while holding a potential of −60 mV (DRG neurons were holding at −60 mV and HEK293 cells were holding at −120 mV). After applying 200-millisecond voltage steps from −70 to +60 mV in +5-mV increments, the current density values were obtained. This allowed analyzing the activation of sodium channels occurring between 0 to 10 mV as a function of voltage through peak current density (normalized to cell capacitance (in picofarads, pF)). For inactivation protocol, clamping at a holding potential of −60 mV, cells were hyperpolarized/repolarized for 1 second from −120 to 0 mV in +10 mV steps. This incremental increase in membrane potential modulates various proportions of sodium channels to the fast inactivated state - in this case the test pulses of 0 mV for 200 milliseconds show fast inactivation when normalized to the maximal sodium current. Glass pipettes with a resistance of 2

to 4 m Ω were used for all recordings. Whole-cell recordings were obtained with a HEKA EPC-10 USB (HEKA Instruments Inc.); data were acquired with a Patchmaster (HEKA) and analyzed with a Fitmaster (HEKA). Capacitive artifacts were fully compensated, and series resistance was compensated by 60 ~ 70%. All experiments were performed at room temperature (at 24 °C-25 °C).

For hERG currents were recorded from HEK293 cells line the internal pipette solution consisted of (in mM): 20 mM KCl, 115 mM K-Aspartic, 1 mM MgCl2, 5 mM EGTA, 10 mM HEPES, 2 mM $Na_2ATP$ (pH 7.4, mOsm/L = 290–310) and external solution consisted of (in mM): 140 NaCl, 3.5 mM KCl, 1 mM $MgCl_2$, 2 mM $CaCl_2$, 10 mM D-Glucose, 10 mM HEPES, 1.25 mM $NaH_2PO4$ (pH 7.4, mOsm/L = 290–310). The hERG-transfected cells were held at -80 mV, and the hERG currents elicited by a depolarization step to +30 mV for 2.5 s before repolarization to −50 mV for 4 s.

## PTX-induced neuropathic pain rat model
Adult male Sprague-Dawley rats were used to investigate PTX-induced neuropathic pain. PTX (2 mg/kg) was administered intraperitoneally every other day for a total of 4 injections (days 0, 2, 4, and 6), resulting in a final cumulative dose of 8 mg/kg. Control animals received an equivalent volume of vehicles. Compounds were administered via Intrathecal injection after 21 days.

## Von Frey test
The von Frey test was utilized to evaluate Mechanical Allodynia in rats. The rats were positioned in a plastic cage furnished with a metal net on the floor and permitted to acclimate for at least an hour prior to each test. A von Frey wire was placed perpendicular to the surface of the paw until it bent. Dixon's "up and down" approach, which involves a progressive increase and decrease in stimulation intensity, was applied to perform the test and collect data[76].

## Data analysis
All values represent the mean ± standard error of the mean (mean ± SEM). Statistical significance of differences between means was determined using Student's $t$ test or one-way ANOVA with Sidak's post hoc test. Von Frey behavioral data sets were analyzed using two-way ANOVA with Sidak's post hoc test. Calcium imaging data were analyzed with MATLAB and compared using $t$ test and two-way ANOVA. Statistical significance was considered for all $P \leq 0.05$. GraphPad Prism 8 was utilized for all graphs.

## Reporting summary
Further information on research design is available in the Nature Portfolio Reporting Summary linked to this article.

## Data availability
All the data is publicly available here: 1. Materials and methods, experimental procedures, computational studies, and bioactivity assessment in the Supplementary Information. 2. The protein data of Human Nav1.7 sodium channel used in this study is available in the PDB database under accession code 5ek0. 3. The OREAL library of the related compounds can be downloaded free of charge via http://www.sysu-sps-compound.com. 4. The source data of Figs. 6–9, Supplementary Fig. 8-10, Table S5, molecular dynamics simulations and virtual screening can be found in the supplementary data 1-10 files.

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

## Acknowledgements

Funding: We are grateful for the support by the National Natural Science Foundation of China (No. 92056201(WH), 82003592 (TS), No.82201376 (SC), No.92256301(WH)). Fundamental Research Funds for the Central Universities, Sun Yat-sen University (No.23ptpy167). Key-Area Research and Development Program of Guangdong Province (No. 2022B1111050003) (WH). Natural Science Foundation of Guangdong Province, Grant No.2021A1515011129 (SC).

## Author contributions

Jirong Shu contributed to computational study, organic synthesis, manuscript preparation and Supplementary Information; Yuwei Wang contributed to bioactivity evaluation, manuscript preparation and Supplementary Information; Weijie Guo and Tao Liu contributed to animal study; Song Cai contributed to concept, manuscript preparation, instructions on experiments, Taoda Shi contributed to concept, manuscript preparation, instructions on experiments; Wenhao Hu contributed to concept and project management.

## Competing interests

The authors declare no competing interests.
