## [Peer Review File · Communications Chemistry]

Reviewers' comments:

Reviewer #1 (Remarks to the Author):

The authors describe a method very similar to V-synthes and used it to develop a novel Nav1.7 inhibitor. While this manuscript reports potentially important work, it is not ready to be published. I list below a number of issues that must be addressed before a decision is made. As a note, I have little expertise in biology and will not discuss these sections.

1. The introduction is not an introduction but rather a summary of the work. A more complete description of existing methods to generate combinatorial libraries, of focused libraries, ... must be provided. As importantly, existing Nav1.7 inhibitors must be discussed (their potency, and issues - why do we need others?). Without this information, I cannot tell whether the computational method led to truly novel inhibitors or if these are simply analogues of existing ones. The summary of the work should be removed and necessary information moved to the method section. Objectives are also missing. What is this work trying to achieve?
2. Some of the terms must be defined. For example, I am not sure many readers would know what a "Bemis-Murcko" structure is. VTD is never defined (vertridine?).
3. It is unclear to what level the method is automated (and usable by medicinal chemists with minimal expertise in computer science). From the reading, it looks like a number of steps were done manually. If so, this should be clearly stated.
4. PCA analysis can hardly be understood unless the descriptors making up the principal components are discussed. Are these descriptors those often used in medicinal chemistry (logP, MW, HBA/HBD,...)? if not, are they meaningful (or likely a consequence of the library rather than capturing medicinal chemistry information)?
5. The manuscript is hard to follow or lacks precision in some places. For example in the section "Virtual screening of Nav1.7 inhibitors generates 42 hits". I would recommend to remind us how many scaffolds, how many compounds per scaffold screened using what method? It would also be interesting to know how much time (real time and cpu time) and how much resources (cpu's and/or gpu's) were required. This would allow us to understand whether this approach is truly much more efficient than HTS. In the section "C4 is the best..." I would recommend to summarize what the experiment is and what it provides. Providing references is putting the burden on readers. What/how is the sodium concentration measured? Can the compounds interfere with this measurement other than through binding to Nav1.7 (e.g., can it chelate sodium)?
6. it is unclear whether the docking was done on the resistant or sensitive channel structure. In fact, until section "C4 is more potent..." there was no mention of the resistance. So I got further confused on what the objective was. Is it to demonstrate the potential of the computational strategy? To develop novel Nav1.7 inhibitors? To inhibit resistant strains? So can docking/MD explain the difference in activity for C4 against wt and mutant?

7. The accuracy of docking/MD is overestimated. "To further confirm the binding mode predicted by molecular docking" Docking and MD CANNOT confirm anything. It can only suggest! Docking programs have shown accuracy of about 50% in several comparative studies (cross-docking) but users keep using these methods as experiments and docking results as fact. This is a major mistake. Docking program developers report much better accuracy but on self-docking (docking small molecules to structure of protein they have been co-crystallized with). So the water molecules and protein conformation are perfectly adjusted to these molecules. In real life, proteins used in prospective studies have been co-crystallized with OTHER small molecules (hence are not adjusted to the ones users are docking). This leads to significantly lower accuracy. The power of MD to identify wrong structure is about as overestimated. If the docked pose is wrong yet good enough, it may stay at least some nanoseconds before leaving, a process therefore not captured by regular MD simulations. This entire MD simulation should either be removed or significantly reduced and revised. All of these RMSD plots should at best be provided as SI, not in the core of the manuscript

Reviewer #2 (Remarks to the Author):

Voltage-gated sodium (Nav) channels are critical for the initiation and the propagation of action potential in excitable cells. Nav1.7 was suggested as a promising target for the development of potential non-addictive analgesics. In this manuscript, Jirong Shu et al used a scaffold-based approach to screen the OREAL library and found several hit compounds that can inhibit Nav1.7, validated by patch-clamp recording of DRG neurons and Nav1.7 expressing HEK293 cells. The potential analgesic effect of C4 was validated in model mice. And MD simulations were performed to reveal possible binding model of C4 in Nav1.7. As requested, this reviewer primarily evaluate the electrophysiological data and structure/function implications because this is the area my expertise is great. The chemistry part will have to be evaluated by others expert reviewers. The manuscript is well written in general. This reviewer has concerns about the logicity in this manuscript and request the authors to provide necessary results to support and strength the conclusions in this study.

Major comments:

1. The best hit compound (C4) was concluded "selectively target Nav1.7" or "selective Nav1.7 inhibitor". Additionally, in abstract, the authors mentioned that C4 has no side effect on cardiac function just because it does not inhibit hERG channel. The evidence to support such conclusions is lacking. The authors should provide related results to support such claims or revise those conclusions.

2. Related to comment 1. In line 198, the results shown in Figure 6 have nothing to do with the selectivity of C4, it just shows C4 can inhibit Nav1.7. To claim subtype selectivity, the authors should validate the effects of C4 on other TTX-S Nav channels of Nav1.1-4, and Nav1.6?

3. In line 177, the subtitle claims C4 is more potent at blocking TTX-R Nav channels than TTX-S Nav channels. Conversely, in lines 182-183, the IC50 values of C4 for TTX-S Nav channels is 40 times lower than that for TTX-R Nav channels. Which one is right? And, where is Figure 5?

4. In lines 154-156, Figure 6A shows example traces of Nav1.7 transiently expressed in HEK293 cell, not the small-diameter rat DRG sensory neurons.

5. How to explain the IC50 value differences of C4 between TTX-S channels in DRG (0.44 μ M) and Nav1.7 transfected HEK293 cells (2.23 μ M)?

6. The authors should explain the rationale for using overnight incubation of C4 in inhibiting Nav1.7 expressing HEK293 cells and using 10 min perfusion of C4 in inhibiting hERG expressing HEK293 cells. Does a 10 min perfusion of C4 also inhibit Nav1.7?

7. This reviewer cannot understand not using the real human Nav1.7 structure (PMID: 30765606; 36424527), rather using a chimera of bacterial NavAb pore-Nav1.7-VSD4 for virtual screening and MD simulations. Do the authors have point mutagenesis results to validate the binding mode of C4 in the docked structure? If C4 inhibits Nav1.7 by binding to VSD4, it should preferentially bind to the activated conformation of VSD4, does C4 display voltage dependent inhibition of Nav1.7?

Other minor comments:

1. In lines 236-256, "5KE0" is not a sodium channel structure.

2. There is no legend for Figure 8, it is hard for reader to know which part of the channel C4 binds to.

Reviewer #3 (Remarks to the Author):

In their publication entitled "Carbenoid-involved Reactions Integrated with Scaffold-based Screening Generates a Nav1.7 Inhibitor" Shu et al describe their efforts to identify a new series of Nav1.7 inhibitors by virtual screening and their cellular and in vivo characterization of the top candidate. As a leader of a

platform company developing new virtual screening approaches I appreciate the challenges of this field. Also, I worked at a company in the past (Genentech) that was heavily focused on the developing Nav1.7 inhibitors, so I appreciate the challenges here as well. Thus I was excited to review this publication to see how this approach was pursued.

Unfortunately, I cannot recommend publication of this submission in its current form. The authors do not provide enough data to validate many of their claims, and the reader is left with more skepticism than learning. I will expand on these concerns in detail.

1. Most importantly, the virtual screening (the main thesis of the paper) does not provide enough information to determine if they have succeeded in their goals. They ultimately only selected and synthesized 42 compounds and identified 10 "hits", for a nearly 25% hit rate. Examples of true virtual screening success with hit rates this high when not based on chemical matter (eg pharmacophore-based) do exist but they are exceptionally rare. So in order to validate it, solid experimental data is needed to convince the reader. No direct protein/ligand interaction data is provided. Such assays do exist in the literature (eg Safina et al, J. Med Chem. 2021, p.2953). In addition, Nav1.X PatchClamp assays are available, and presenting a panel of the various Navs would have increased confidence if selectivity was observed. In the end, all the data can be characterized as phenotypic, which are not sufficient to validate a virtual screen. In particular this is challenging since so few details are provided about the virtual screen itself.

2. The chemical matter itself is concerning. The dihydro-oxazole with the exocyclic methylene has a lot of options for reactivity. Such a core is not well known in the medicinal chemistry literature, and given the reactivity concerns there is a heightened reason for skepticism of the direct effect predicted by the virtual screen.

3. Even if the in vitro data were solid, the in vivo data also presents many reasons for concern. No dose of C4 is given for the efficacy study, but a compound with a micromolar level cellular function would be very challenging to see on-target in vivo activity. For example, the Genentech compounds (see reference above) show in vivo activity when they exhibit low nanomolar cellular function. It is possible that the two series are differentiated by plasma protein binding to make this possible, but without PPB data nor in vivo blood/plasma levels of C4 the reader is left with even further reasons for skepticism.

There are a few additional minor/typographical issues which I will call out which on their own are not reason for rejection, but must be corrected if the other issues are addressed. These are:

1. The authors claim that external libraries are in the 100s of millions in the first paragraph. Enamine REAL is now up to 10s of billions.

2. The images in Figure 3 are too small to be useable. In particular the graph in the bottom right corner.

3. Figure 5 seems to be missing altogether.

Dear reviewers,

Thank you for your insightful and constructive feedback. We have revised the manuscript according to your advice. The point-by-point response to your questions are as follows:

Reviewer 1

1. The introduction is not an introduction but rather a summary of the work. A more complete description of existing methods to generate combinatorial libraries, of focused libraries, ... must be provided. As importantly, existing Nav1.7 inhibitors must be discussed (their potency, and issues - why do we need others?). Without this information, I cannot tell whether the computational method led to truly novel inhibitors or if these are simply analogues of existing ones. The summary of the work should be removed, and necessary information moved to the method section. Objectives are also missing. What is this work trying to achieve?

Answer: Thank you for your constructive and helpful suggestions. We have rewritten the introduction according to your advice. We believe the new introduction can help our future read make a judgement on the novelty of the Nav1.7 inhibitor.

2. Some of the terms must be defined. For example, I am not sure many readers would know what a "Bemis-Murcko" structure is. VTD is never defined (vertridine?).

Answer: Thank you for your guidance, we have added definitions of the corresponding terms in the manuscript.

Bemis Murdo structure is used for analyzing the shape and structure of drugs. This method decomposes molecules into four types of units: ring system, linker, scaffold, and side chain. Bemis Murdo structure is represented by scaffold, which is composed of only ring system and linker, while side chains are removed from drug molecules.

Veratridine (VTD) is a voltage gated sodium channel (VGSC) regulator used as an agonist in the functional screening of VGSC blockers. By using the VTD-evoked calcium imaging assay, the inhibition rate of compounds on sodium channels can be indirectly measured.

3. It is unclear to what level the method is automated (and usable by medicinal chemists with minimal expertise in computer science). From the reading, it looks like a few steps were done

manually. If so, this should be clearly stated.

Answer: Thank you for your concern about the automation of the method. Library generation and the molecular docking was done with the aid of automated tools, while the characterization of reaction scope, synthesis of molecules and evaluation of predicted activity was done manually. With your requirement of automation in mind, we are planning to train AI models to characterize the reaction scope and generate the corresponding library.

4. PCA analysis can hardly be understood unless the descriptors making up the principal components are discussed. Are these descriptors those often used in medicinal chemistry (logP, MW, HBA/HBD,...)? if not, are they meaningful (or likely a consequence of the library rather than capturing medicinal chemistry information)?

Answer: Thank you for your kind guidance. We have added the following PCA introduction in the manuscript. "PCA analysis is a dimensionality reduction algorithm that can reduce multiple variables on a two-dimensional plane for visual analysis. Through PCA analysis, the chemical space of OBL compound libraries and other compound libraries with sodium ion channel inhibitory activity can be visualized. By calculating the various chemical properties of compounds in different compound libraries, which are often related to compound's druggability, and analyzing them through dimensionality reduction algorithms, the chemical space is drawn based on molecular properties, and finally the chemical spaces of different compound libraries are compared."

5. The manuscript is hard to follow or lacks precision in some places. For example in the section "Virtual screening of NaV1.7 inhibitors generates 42 hits". I would recommend to remind us how many scaffolds, how many compounds per scaffold screened using what method? It would also be interesting to know how much time (real time and cpu time) and how much resources (cpu's and/or gpu's) were required. This would allow us to understand whether this approach is truly much more efficient than HTS.

Answer: Thank you for your concern on the details of screening. We used 1278 representative compounds from OREAL compound library for docking calculations. Typically, a scaffold compound corresponds to 200-1000 molecules (depending on the reaction boundary of each Bemis Murdo structure's related literature). Due to the experimental conditions, we did not use the original V-synthesis code, but instead followed its strategy of screening, that is screening the skeleton at first, and then exploring the influence of substituents. We do not have specific statistics on the resources consumed.

6. In the section "C4 is the best..." I would recommend to summarize what the experiment is and what it provides. Providing references is putting the burden on readers. What/how is the sodium concentration measured? Can the compounds interfere with this measurement other

than through binding to Nav1.7 (e.g., can it chelate sodium)?

Answer: Thank you for your kind questions about the measurement of compound C4 and for guiding us to better communicate with our readers. We apologize for not explaining the methods to our readers in detail. The following is the description of the methods we used: "Calcium imaging" is a technique employed to investigate fluctuations in intracellular calcium ion concentrations within cells. Calcium ions play pivotal roles in cellular signaling and regulatory functions, and understanding their dynamic changes is essential for comprehending both cellular physiology and pathological processes. We utilize the fluorescent dye Fluo-4am to probe cellular calcium ion concentrations. By pre-incubating the dye to facilitate its penetration into cells and subsequent binding to calcium, fluorescence is induced. Upon application of a sodium ion channel agonist to stimulate cells, the activation of sodium channels indirectly results in the release of intracellular calcium ions, generating fluorescence changes. We employ fluorescence microscopy to monitor and record these changes, thereby exploring the effects of different compounds on sodium ion channels and elucidating any distinctions. The output obtained through this method is the fluorescence change ratio. By comparing the magnitude of fluorescence changes, we can indirectly and efficiently assess the impact of compounds on sodium ion channels. However, specific sodium ion concentration values cannot be directly obtained. Through experimentation with a series of compounds using this approach, compound C4 exhibited the most pronounced fluorescence change, indicating the most effective inhibitory effect. As compounds do not chelate with sodium ions, they bind to sodium channels, which function as channel proteins, akin to compounds binding to target proteins.

6. it is unclear whether the docking was done on the resistant or sensitive channel structure. In fact, until section "C4 is more potent..." there was no mention of the resistance. So I got further confused on what the objective was. Is it to demonstrate the potential of the computational strategy? To develop novel Nav1.7 inhibitors? To inhibit resistant strains? So can docking/MD explain the difference in activity for C4 against wt and mutant?

Answer: Thank you for bringing up the objective question again, which is the central issue of our previous manuscript. Under your helpful guidance, we rewrote the introduction section and made our objective clear: we aim to develop novel Nav1.7 inhibitors by using a newly developed screening library. Meanwhile, we borrowed the advanced screening strategy from V-SYNTHESIS.

Upon our goal of targeting Nav1.7, we planned to look for inhibitors of wildtype Nav1.7 at first and use them to deal with the acquired pain conditions like chemotherapy-induced chronic pain. In these pain conditions, WT Nav1.7 is usually overexpressed. There are also inherited pain disorders caused by the mutated Nav1.7 as you mentioned (*Nat. Rev. Neurosci.*, 2013, 14, 49-62). For example, S456X, I767X, W897X (X stands

for) mutants lead to the inactivation of Nav1.7 and cause indifference to pain, while E406K over-activates Nav1.7 and causes erythromelalgia (*Nature*, 2006, 444, 894-898). In future study, we will investigate the potential of C4 and its derivatives in inhibiting these mutants.

The following is our explanation of the features of C4 as a Nav1.7 inhibitor. Sodium ion channels, membrane proteins responsible for regulating the influx and efflux of sodium ions across the cell membrane, exhibit various subtypes classified based on their sensitivity to specific toxins, notably tetrodotoxin (TTX). TTX is a toxin that selectively blocks the function of sodium ion channels. This categorization primarily hinges on sensitivity to TTX, dividing sodium ion channels into two classes: 1. TTX-S (Tetrodotoxin-Sensitive): Sodium ion channels sensitive to TTX. Notable subtypes within this class encompass Nav1.1, Nav1.2, Nav1.3, Nav1.4, Nav1.6, Nav1.7, and Nav1.8, with Nav1.7 representing a predominant subtype. These channels play critical roles in neurons and muscle cells, participating in processes such as action potential propagation and neuromuscular transmission. 2. TTX-R (Tetrodotoxin-Resistant): Sodium ion channels relatively insensitive to TTX. This category includes subtypes such as Nav1.5 and Nav1.9. Typically found in cardiac cells and certain sensory neurons, these channels are associated with processes like conduction speed and neurotransmitter release. In our biological experiments, we employ TTX to validate whether compound C4 belongs to the TTX-S category. If C4 exhibits a more pronounced inhibition of TTX-S channels, it suggests a greater inhibitory potential against TTX-S, prompting further investigation into the specific binding of C4 to Nav1.7 within TTX-S.

Upon the question "Can docking/MD explain the WT explain the difference in activity for C4 against wt and mutant, molecular docking/MD may not be able to give a convincing explanation to the activity difference between WT and mutants, since the allosteric site of these two proteins are strikingly different. The following are the details of the difference:

Take inherited erythromelgia (IEM) relative mutation as example (J Gen Physiol 4 December 2023; 155 (12): e202313450), there are striking differences between wt and different mutants.

The picture shows the NaV1.7 S211P/NaVAb E96P(PDB: 8DIW) mutation, which is different from the wt NaVAb(PDB: 3RVY) and the protein current used for docking (PDB: 5EK0) . In the allosteric site, the original binding cavity disappears in various allosteric variants. This poses a challenge to obtaining the difference between wt and mt through a simple docking/MD method. In the subsequent work, we will refer to your suggestions and examine the specific site results.

7. The accuracy of docking/MD is overestimated. "To further confirm the binding mode predicted by molecular docking" Docking and MD CANNOT confirm anything. It can only suggest! Docking programs have shown accuracy of about 50% in several comparative studies (cross-docking) but users keep using these methods as experiments and docking results as fact. This is a major mistake. Docking program developers report much better accuracy but on self-docking (docking small molecules to structure of protein they have been co-crystallized with). So, the water molecules and protein conformation are perfectly adjusted to these molecules. In real life, proteins used in prospective studies have been co-crystallized with OTHER small molecules (hence are not adjusted to the ones users are docking). This leads to significantly lower accuracy. The power of MD to identify wrong structure is about as

overestimated. If the docked pose is wrong yet good enough, it may stay at least some nanoseconds before leaving, a process therefore not captured by regular MD simulations. This entire MD simulation should either be removed or significantly reduced and revised. All of these RMSD plots should at best be provided as SI, not in the core of the manuscript.

Answer: Thank you for your guidance on the molecular docking/MD section, we have removed the MD section from the manuscript to supplementary material according to your advice.

Reviewer 2

Major comments:

1. The best hit compound (C4) was concluded “selectively target Nav1.7” or “selective Nav1.7 inhibitor”. Additionally, in abstract, the authors mentioned that C4 has no side effect on cardiac function just because it does not inhibit hERG channel. The evidence to support such conclusions is lacking. The authors should provide related results to support such claims or revise those conclusions.

Answer: Thank you for your correction. We apologize for the unsafe conclusion on the “no side effect on cardiac function” We have change” Preliminary toxicology studies showed that C4 is negative to hERG, implying that C4 has no effect on cardiac function.” to “The preliminary toxicology studies indicate that under acute conditions, compound C4 does not affect the hERG channel. This finding holds promise in mitigating potential adverse effects on cardiac electrophysiology during the acute phase of drug administration. However, this represents only one component of the cardiac safety assessment, and further in-depth research and validation are required to explore other potential adverse impacts. Nevertheless, this study provides valuable preliminary insights into the cardiac safety assessment of our compound”.

2. Related to comment 1. In line 198, the results shown in Figure 6 have nothing to do with the selectivity of C4, it just shows C4 can inhibit Nav1.7. To claim subtype selectivity, the authors should validate the effects of C4 on other TTX-S Nav channels of Nav1.1-4, and Nav1.6?

Answer: Thank you for your concern on the subtype selectivity. We conducted additional experiments on the effects of compound C4 on Nav1.1-Nav1.4, Nav1.6 subtypes, their inhibition rates at 20 μ M are as follows: 8.63% \pm 0.73%, 9.24% \pm 5.10%, 27.94% \pm 1.18%, 16.02% \pm 0.04%, and 7.71% \pm 1.19%, respectively. Compared to around 85% inhibition against Nav1.7 (Figure 8C and D), C4 showed decent subtype selectivity toward Nav1.7. These findings have been documented in the supplementary materials.

3. In line 177, the subtitle claims C4 is more potent at blocking TTX-R Nav channels than TTX-S Nav channels. Conversely, in lines 182-183, the IC₅₀ values of C4 for TTX-S Nav channels is

40 times lower than that for TTX-R Nav channels. Which one is right? And where is Figure 5?

Answer: Thank you for correcting our mistakes. We have made the change: “C4 is more potent against tetrodotoxin-sensitive Nav Channels than against tetrodotoxin-resistant Nav channels of DRG Neurons.” And, we added figure 5 back to the manuscript. We apologize for the carelessness.

4. In lines 154-156, Figure 6A shows example traces of Nav1.7 transiently expressed in HEK293 cell, not the small-diameter rat DRG sensory neurons.

Answer: Thank you for your inquiry. I acknowledge the errors in the descriptions of Figures 4 and 6 in our previous submission. In the revised version, we have ensured that the images and their descriptions are correctly matched.

5. How to explain the IC₅₀ value differences of C4 between TTX-S channels in DRG (0.44 μM) and Nav1.7 transfected HEK293 cells (2.23 μM)?

Answer: Thank you for your question about the IC₅₀ difference between TTX-S channels in DRG and Nav1.7 transfected HEK293 cells. In the dorsal root ganglia (DRG) cells located in the spinal cord segment, various Tetrodotoxin-Sensitive (TTX-S) receptor subtypes exist, including Nav1.1, Nav1.6, Nav1.7, Nav1.8, and Nav1.9. Among these, Nav1.7 is a prevalent subtype, though it is not the sole subtype expressed. Notably, the expression levels of these receptor subtypes, particularly Nav1.7, can vary not only between different cell types within the same tissue, such as DRG cells, but also among individual cells within the same cell population. Even in the case of HEK293 cells transfected with Nav1.7, the predominant expression is of the Nav1.7 subtype, though the concentrations may not be entirely equivalent. The distribution of Nav1.7 receptors on the cell membrane also exhibits variability among individual DRG cells, leading to differences in receptor numbers. Consequently, replicating an experiment may yield non-identical results, yet these variations collectively contribute to an overall quantitative range. Additionally, the range of errors observed in the experiment falls within our acceptable limits.

6. The authors should explain the rationale for using overnight incubation of C4 in inhibiting Nav1.7 expressing HEK293 cells and using 10 min perfusion of C4 in inhibiting hERG expressing HEK293 cells. Does a 10 min perfusion of C4 also inhibit Nav1.7?

Thank you for your question. In toxicity assessments, hERG (human Ether-à-go-go-Related Gene) experiments primarily focus on the acute cardiac toxicity of drugs, especially their effects on cardiac electrophysiology. The hERG channel is a crucial potassium channel in cardiac cells, essential for maintaining normal electrophysiological functions of the heart. Drug-induced inhibition of the hERG channel may lead to prolonged QT intervals, an electrocardiographic change that can result in severe arrhythmias, such as Torsades de Pointes. Therefore, hERG experiments are generally

considered an important part of assessing a drug's acute cardiac toxicity rather than chronic toxicity evaluations. Thus, we utilize an acute perfusion method to assess the impact of compounds on the hERG channel.

Additionally, to evaluate the acute effects on Nav1.7, we measured the impact of the compound at intervals of acute (10 minutes), three hours, and overnight. The observed inhibition rates of the compound on Nav1.7 at these respective time points were $30.4\% \pm 7.37\%$, $30.5\% \pm 8.56\%$, and $63.4\% \pm 2.39\%$ (as illustrated in the figure below). These findings suggest that compound C4 exerts an inhibitory effect on Nav1.7 under acute conditions. This data has been included in the supplementary materials.

7. This reviewer cannot understand not using the real human Nav1.7 structure (PMID: 30765606; 36424527), rather using a chimera of bacterial NavAb pore-Nav1.7-VSD4 for virtual screening and MD simulations. Do the authors have point mutagenesis results to validate the binding mode of C4 in the docked structure? If C4 inhibits Nav1.7 by binding to VSD4, it should preferentially bind to the activated conformation of VSD4, does C4 display voltage dependent inhibition of Nav1.7?

Answer: thank you for your insightful questions on the Nav1.7 protein we used for screening. The goal of screening is to look for inhibitors binding to the allosteric site at VSD_{IV} domain of Nav1.7. We chose bacterial NavAb pore-Nav1.7-VSD4 (5ek0) for two reasons: 1) Bacterial NavAb pore-Nav1.7-VSD4 protein is structurally reserved with mammalian (Science350, aac5464(2015). J Gen Physiol 4 December 2023; 155 (12): e202313450); 2) It is the co-crystallized protein with the GX-936 which is a proven allosteric inhibitor. Currently, 5EK0 is a commonly used protein crystal for molecular docking of VSDIV binding domain inhibitors of Nav1.7 (J. BIOMOL. STRUCT DYN., 2021,39, 4472-4479; J. Med. Chem. 2019, 62, 2, 908-927; Comput. Biol. Chem, 2018,77, 214-225; J. Med. Chem.2019, 62, 4091-4109). 2) The recommended 6J8G and 6J8H for Nav1.7-HS and 6J8I and 6J8J for Nav1.7-PT are not appropriate in our docking approach because the toxin binds to the VSDIV binding domain of these proteins, which can affect ligand docking result.

Upon the binding mode of C4, our data showed that C4 accelerates the inactivation of sodium channels while decelerating their activation, which is consistent with the mechanism of VSDIV-targeted inhibitors (Figure 6D and 6E). Plus, as you suggested, voltage-dependent inhibition experiment will further validate the VSDIV binding, to test this out, we observed C4 showed voltage-dependent inhibition at 0.1 μM (as shown in the figure below). As for point mutagenesis, which is also another solid tool for validating the binding mode, we haven't collected data yet. We totally agree with you on the importance of the tool, and we are still building the point mutagenesis platform for future study.

Other minor comments:

1. In lines 236-256, "5KE0" is not a sodium channel structure.

Answer: Thank you for your careful correction. The mistake has been fixed.

2. There is no legend for Figure 8, it is hard for reader to know which part of the channel C4 binds to.

Answer: Thank you for your suggestion. The legend of Figure 8 has been added.

Reviewer 3

1. Most importantly, the virtual screening (the main thesis of the paper) does not provide enough information to determine if they have succeeded in their goals. They ultimately only selected and synthesized 42 compounds and identified 10 "hits", for a nearly 25% hit rate. Examples of true virtual screening success with hit rates this high when not based on chemical matter (eg pharmacophore-based) do exist but they are exceptionally rare. So in order to validate it, solid experimental data is needed to convince the reader. No direct protein/ligand interaction data is provided. Such assays do exist in the literature (eg Safina et al, J. Med Chem. 2021, p.2953). In addition, Nav1.X PatchClamp assays are

available, and presenting a panel of the various Navs would have increased confidence if selectivity was observed. In the end, all the data can be characterized as phenotypic, which are not sufficient to validate a virtual screen. In particular this is challenging since so few details are provided about the virtual screen itself.

Answer: Thank you for your concern on the hit rate. We increased the threshold of the screening data, from 20% to 50%. And “50%” is the real number we used in action, and the number “20%” is misused in our previous manuscript. We apologize for the miscommunication. And we added the detailed scoring results to the supplementary material.

Upon the validation of the direct binding of C4 and Nav1.7, we thank you for providing the assay of isotope-labeled ligand exchange. However, when we chased down to the paper (J. Med Chem. 2021, p.2953), we found that the isotope-labeled ligand is not commercially available in China. Then we checked if we could synthesize the ligand by ourselves, but the labeled atom was not revealed in the paper. Meanwhile, the radioactive reagents budget reaches its limit in our school according to the regulations of radioactive reagents. We totally agree with you on the significance of direct binding validation. However, the current condition does not allow us to do this. Actually, lacking convenient and straightforward method of validating direct binding is one of a general issue in the Nav1.7 inhibitor field.

Upon the subtype selectivity, we took your advice to run Nav1.X PatchClamp assays against Nav1.1-1.4 and Nav1.6. Their inhibition rates at 20 μ M drug concentration are as follows: 8.63% \pm 0.73%, 9.24% \pm 5.10%, 27.94% \pm 1.18%, 16.02% \pm 0.04%, and 7.71% \pm 1.19%. In terms of inhibition rate at 20 μ M, the selectivity index towards Nav1.7 are around 7, 7, 2, 4, and 8-fold, respectively. C4 shows subtype selectivity to some extent. However, compared to the known potent VSDIV inhibitor which is of around 1000-fold subtype selectivity, C4 is still at its early stage and gets far way to go at this aspect. We assume the following medicinal chemistry optimization effort will further improve the selectivity index.

Referring to the screening details, we added the scores and structures of hits to supplementary material.

2. The chemical matter itself is concerning. The dihydro-oxazole with the exocyclic methylene has a lot of options for reactivity. Such a core is not well known in the medicinal chemistry literature and given the reactivity concerns there is a heightened reason for skepticism of the direct effect predicted by the virtual screen.

Answer: Thank you for your concern on “the exocyclic methylene moiety”. According to our experience of handling compound C4, it is stable for months at room temperature. But indeed, we cannot 100% exclude the possibility of “off-target effect” in cells, due to exocyclic methylene moiety in 4,5-dihydrooxazole, C4 is of a possibility to react with

cellular strong nucleophiles like thiols and aminos. Currently, we don't observe severe side effects in our experiments. In future medicinal chemistry optimization, we will take your concern on the "exocyclic methylene moiety" into consideration to perform the iteration of this series of compounds.

The fact that C4 shows some subtype selectivity towards Nav1.7 indicates C4 is on the target. Still, we must admit that we are not 100% sure that C4 directly binds to Nav1.7 until we can detect the direct binding event through the method that we are able to access to.

3. Even if the *in vitro* data were solid, the *in vivo* data also presents many reasons for concern. No dose of C4 is given for the efficacy study, but a compound with a micromolar level cellular function would be very challenging to see on-target *in vivo* activity. For example, the Genentech compounds (see reference above) show *in vivo* activity when they exhibit low nanomolar cellular function. It is possible that the two series are differentiated by plasma protein binding to make this possible, but without PPB data nor *in vivo* blood/plasma levels of C4 the reader is left with even further reasons for skepticism.

Answer: Thank you for your concern on the *in vivo* data. Firstly, we apologize for not giving the dose of C4 in our previous manuscript. Now the dose 20ug/10ul (concentration by local administration) has been added to the experiment details. Your prediction on the difference of PPB or of *in vivo* blood/plasma level is critical and we will pay attention to it in the coming hit-to-lead/candidate optimization, especially for systematic injection. As for the local administration in our current *in vivo* experiment, the impact of PPB is expected to be less than the systematic injection.

There are a few additional minor/typographical issues which I will call out which on their own are not reason for rejection but must be corrected if the other issues are addressed. These are:

1. The authors claim that external libraries are in the 100s of millions in the first paragraph. Enamine REAL is now up to 10s of billions.

Answer: Thank you for comparing our library to Enamine REAL (We feel honored!). In terms of scale, it's a fact that these two libraries are totally at two different levels. If our library is currently a small lake, Enamine REAL is a vast sea. Actually, Enamine REAL inspired us to use our unique chemistry to build a unique library. What we want to express to the public is that we built a library that could expand the chemical space for drug discovery and find an interesting molecule for stopping pain from the library, and be complementary to Enamine REAL.

2. The images in Figure 3 are too small to be useable. In particular the graph in the bottom right corner.

Answer: Thank you for suggestions on Figure 3. We have separated the figure into 2 figures (Figure 3 and figure 4) and they are clear and readable.

3. Figure 5 seems to be missing altogether

Thank you for your careful and kind correction. We have added figure 5 to the manuscript.

Again, please allow us to thank you for the informative, insightful, and constructive comments and suggestions. We believe the revised manuscript is much improved and hope to reach the quality of publication in Communications Chemistry.

Yours sincerely

Song Cai, Taoda Shi and Wenhao Hu

蔡松 史滔达 胡皓

REVIEWERS' COMMENTS:

Reviewer #1 (Remarks to the Author):

I would like to thank the authors for addressing some of my concerns. However, some remain:

1. The introduction still contains sections that are not expected in an introduction. In particular, the description of the work (Figures 2 and 3 and associated text) should be part of the result and discussion (or a methodology section) not of the introduction
2. The level of automation of the protocol should be discussed in the manuscript.
3. My point about the PCA was misunderstood. This dimensionality reduction comes with some PC's being combinations of original descriptors. My question was which descriptors are incorporated into these PC's and what are their weights. With this information, medicinal chemists will be able to get a better sense of the importance of some properties over others.

Minor comments:

1. Bemis Murcko became Bemis Murdo

Reviewer #2 (Remarks to the Author):

The manuscript has been improved. The authors addressed some of my previous concerns. However, I have two concerns remaining.

1. Thanks for showing the inhibition of C4 on Nav1.7 on different incubation time points. It is hard to understand why there is no difference between acute (how long exactly?) and 3-hour points with an inhibition rate of ~30%, but increased to 63% when incubation overnight.
2. The authors explained why using a Nav1.7-VSD4-NavAb chimera as a model for docking, and listed four related studies. However, three of them were published before the publication of human Nav1.7 structures. I would suggest the authors at least appreciate the availability of human Nav1.7 structures in complex with different small-molecule inhibitors. The author hastily claimed those Nav1.7 structures with toxins bound to the VSD4 is not accurate!

Reviewer #3 (Remarks to the Author):

I appreciate the authors taking the time to carefully address all the reviewers comments. In particular, attempting to address the on-target criticisms by doing patchclamp assays of the other NaV channels could have been informative. I am confused, however, why this data is not included in the revised publication but rather is just provided as a comment to this reviewer's initial review. Further, the data (even in the comment) is only provided as a single point concentration, so it is not possible to conclude much about selectivity from this data.

I am also confused why the authors commented on the dose of the inhibitor for the in vivo study, but this still did not include this in the publication. I am representing the readership (not myself!) when I share that this is important information for the reader. Also, the comment in the rebuttal letter that PPB is less important because the delivery is "local" is not correct. The CNS is full of protein, and brain binding can be even higher than plasma.

I am still very skeptical that these inhibitors are on-target. But I take the author's rebuttal that definitive proof (via the Genentech binding assay, for example) is outside the scope of what they can achieve. If they correct the two issues above (including off-target PatchClamp data, ideally as full DRCs; and including the in vivo dose) I would be supportive of publication.

Dear reviewers,

Thank you heartfully again for your insightful questions and constructive suggestions. With your professional assistance, we believe the manuscript is obviously improved and close to the level of publishing in Communications Chemistry. The following is our 2nd round of response to your questions point-by-point.

Reviewer #1 (Remarks to the Author):

I would like to thank the authors for addressing some of my concerns. However, some remain:

1. The introduction still contains sections that are not expected in an introduction. In particular, the description of the work (Figures 2 and 3 and associated text) should be part of the result and discussion (or a methodology section) not of the introduction

Thank you for your suggestion. We have referred to your suggestion and transferred some of the content of Figure 3 to the results and discussion sections for introduction.

2. The level of automation of the protocol should be discussed in the manuscript.

Thank you for your suggestion. We have added an introduction to the automation level of the screening process in the main text. Overall, we have not been able to achieve automated processes between each step. Within each step, except for manual confirmation of results, compound library generation, information search, and virtual high-throughput screening are carried out based on open-source chemical informatics data packages and commercial docking software. We have added description of automation level in line 100.

3. My point about the PCA was misunderstood. This dimensionality reduction comes with some PC's being combinations of original descriptors. My question was which descriptors are incorporated into these PC's and what are their weights. With this information, medicinal chemists will be able to get a better sense of the importance of some properties over others.

Thank you for your suggestion. We apologize for misunderstanding your meaning. The caption of Figure 5 in the original text clearly states the content of the principal components we used, and the contribution of the principal components is depicted in the SI section. This is indeed not conducive to your and readers' reading. Therefore, we have added a contribution table of the original indicators to PC1 and PC2 in the SI section for readers to better

understand the relevant information.

Minor comments:

1. Bemis Murcko has been changed to Bemis Murdo

Reviewer #2 (Remarks to the Author):

The manuscript has been improved. The authors addressed some of my previous concerns. However, I have two concerns remaining.

1. Thanks for showing the inhibition of C4 on Nav1.7 on different incubation time points. It is hard to understand why there is no difference between acute (how long exactly?) and 3-hour points with an inhibition rate of ~30%, but increased to 63% when incubation overnight.

Thank you for your concern regarding the interesting observation. Honestly, without information provided by structural biology, we can't give a convincing explanation on the phenomenon. But we would like to offer a hypothesis on this. C4 may bind to multiple sites on Nav1.7, and the sequential binding events possibly show positive cooperativity. The first binding event helps the followed binding event. But we are not sure how long it takes to transfer the conformation change between the binding events, and whether the kinetics is consistent with the observation in patch clamp assay. These intriguing questions will take more effort to answer.

2. The authors explained why using a Nav1.7-VSD4-NavAb chimera as a model for docking and listed four related studies. However, three of them were published before the publication of human Nav1.7 structures. I would suggest the authors at least appreciate the availability of human Nav1.7 structures in complex with different small-molecule inhibitors. The author hastily claimed those Nav1.7 structures with toxins bound to the VSD4 is not accurate!

Thank you for pointing out our mistake when we replied to you last time. We totally agree with you on the availability of human Nav1.7 structures in complex with different small-molecule inhibitors. In our last reply, what we wanted to express was that the binding of relevant toxins may affect the molecular docking results of compounds in the VSD4 binding domain. TTX itself acts as pore blockers, which physically occludes the channel pore, while HWTX-IV and ProTx-II act as gating modifier toxins binding to the position of VSDII, binding to the peripheral region that links S3 and S4.

We also attempted to refer to some of the latest reports on small molecule inhibitors (Nat Struct Mol Biol 29, 1208-1216 (2022)). These compounds are

classic inhibitor that can guide us well in virtual screening on Nav1.7 inhibitors. However, the binding sites of these crystals (7XM9, 7XMF, 7XMG) are in the PD region and belong to the category of pore blockers, which is different from the allosteric binding sites we want to screen. Therefore, it was not continued to be used in the end. However, when the human crystal containing PF-05089771 paper was online (PDB: 8I5G, Nat Commun. 14, 3224 (2023)), we had already completed the work and was preparing manuscript. We are excited to see the high quality cocrystal structures published and we believe this will improve our future structure-based drug discovery.

Reviewer #3 (Remarks to the Author):

I appreciate the authors taking the time to carefully address all the reviewers' comments. In particular, attempting to address the on-target criticisms by doing patchclamp assays of the other NaV channels could have been informative. I am confused, however, why this data is not included in the revised publication but rather is just provided as a comment to this reviewer's initial review. Further, the data (even in the comment) is only provided as a single point concentration, so it is not possible to conclude much about selectivity from this data.

Thank you for your concern on the subtype selectivity issue. We have added the discussion in the main text. Upon the dose, we appreciate your carefulness about the dose issue. The IC_{50} value of C4 against Nav1.7 is around $2.3 \mu M$ and it gives around 90% inhibition at $10 \mu M$. To save cost of experiment, we therefore choose a single $20 \mu M$ dose to test against the other subtypes, resulting in less than 20% inhibition. The experiment was conducted in triplicate. Thus, we hold confidence in the observed subtype-selectivity.

I am also confused why the authors commented on the dose of the inhibitor for the in vivo study, but this still did not include this in the publication. I am representing the readership (not myself!) when I share that this is important information for the reader. Also, the comment in the rebuttal letter that PPB is less important because the delivery is "local" is not correct. The CNS is full of protein, and brain binding can be even higher than plasma.

I am still very skeptical that these inhibitors are on-target. But I take the author's rebuttal that definitive proof (via the Genentech binding assay, for example) is outside the scope of what they can achieve. If they correct the two issues above (including off-target Patch Clamp data, ideally as full DRCs; and including the in vivo dose) I would be supportive of publication.

Thank you again for bringing up questions for our future readers. **We added dose of C4 for intrathecal injection in the Result section (C4 effectively**

reverses PINP)(line 274).

Upon your concern on the Plasma protein binding (PPB) issue, we agree with you on the fact PPB events exist in CNS as well. However, PPB remains a controversial topic in drug discovery (Expert Opinion on Drug Discovery, 2021, 16, 1453-1465). The following are the opinions we quote from the XODD paper: *"Many successful drugs are highly plasma protein bound with PPB > 99% (Smith DA, Di L, Kerns EH. **The effect of plasma protein binding on in vivo efficacy: misconceptions in drug discovery.** Nat Rev Drug Discovery. 2010;9(12): 929–939.). PPB is a parameter that generally cannot be used to differentiate compounds in terms of their effectiveness in vivo.*

Thank you again for your kind suggestions on confirming the PPB issue. In the case of compound **C4** which is still at the stage of hit, we gave the priority to its in vivo efficacy. We will take PPB factor into consideration in our hit-to-lead optimization.